# Global Context-aware Representation Learning for Spatially Resolved Transcriptomics

Yunhak Oh [* 1]   Junseok Lee [* 2]   Yeongmin Kim [3]   Sangwoo Seo [2]   Namkyeong Lee [2]   Chanyoung Park [1 2]

## Abstract

Spatially Resolved Transcriptomics (SRT) is a cutting-edge technique that captures the spatial context of cells within tissues, enabling the study of complex biological networks. Recent graph-based methods leverage both gene expression and spatial information to identify relevant spatial domains. However, these approaches fall short in obtaining meaningful spot representations, especially for spots near spatial domain boundaries, as they heavily emphasize adjacent spots that have minimal feature differences from an anchor node. To address this, we propose Spotscape, a novel framework that introduces the Similarity Telescope module to capture global relationships between multiple spots. Additionally, we propose a similarity scaling strategy to regulate the distances between intra- and inter-slice spots, facilitating effective multi-slice integration. Extensive experiments demonstrate the superiority of Spotscape in various downstream tasks, including single-slice and multi-slice scenarios. Our code is available at the following link: https://github.com/yunhak0/Spotscape.

## 1. Introduction

Recently, Spatially Resolved Transcriptomics (SRT) has gained significant attention for its ability to capture the spatial context of cells within tissues. It provides spatially resolved gene expressions, quantifying gene activity levels and mapping each spot's physical location within the tissue. Although it helps uncover complex transcriptional structures in tissue, analyzing SRT data remains challenging due to

noise caused by the technology's limited resolution and high dimensionality.

In response to these challenges, representation learning methods have been developed to capture biologically meaningful spot representations by integrating spatial and gene expression data. Specifically, graph-based methods (Xu et al., 2024a; Hu et al., 2021) construct graphs using spatial coordinates to gather information from nearby spots and generate representations using graph neural networks (GNNs). While this approach effectively incorporates spatial information into latent representations, it faces limitations for spots near spatial domain boundaries. These boundary spots may receive information from nodes representing different types of spots (i.e., heterophilic nodes), which can complicate accurate representation learning. To address this limitation, STAGATE (Dong & Zhang, 2022) utilized graph attention networks (GAT) (Veličković et al., 2017) to learn spot similarities, enhancing the representations of spots at spatial domain boundaries.

Despite the effectiveness of STAGATE, we argue that learning attention weights in SRT data remains challenging due to the *continuous nature* of biological systems, where gene expression values vary smoothly along spatial coordinates (Cembrowski & Menon, 2018; Phillips et al., 2019; Adler et al., 2019; Harris et al., 2021). This continuity blurs the distinction between spatial domains, as shown in Figure 1 (a), since spots in local neighborhoods exhibit high similarity even across different domains. Furthermore, even with well-learned edge weights (e.g., assigning high weights to same-type spots and low weights otherwise), an anchor spot may still struggle to extract meaningful information from its neighbors due to marginal feature differences. As a result, the received information may be largely redundant, hindering representation quality. To corroborate our argument, we compared the clustering performance of various graph autoencoder (GAE) architectures, as shown in Figure 1 (b): (1) GAE on the original spatial nearest neighbor (SNN) graph[1], (2) GAE with a GAT encoder, (3) GAE with oracle

*Equal contribution [1]Graduate School of Data Science, KAIST, Daejeon, Republic of Korea [2]Department of Industrial & Systems Engineering, KAIST, Daejeon, Republic of Korea [3]Department of Mathematical Sciences, KAIST, Daejeon, Republic of Korea. Correspondence to: Chanyoung Park <cy.park@kaist.ac.kr>.

*Proceedings of the 42nd International Conference on Machine Learning*, Vancouver, Canada. PMLR 267, 2025. Copyright 2025 by the author(s).

---

[1]The SNN graph is constructed by connecting spots that are either within a predefined radius $r$ or among the nearest top $k$ neighbors based on spatial distance.

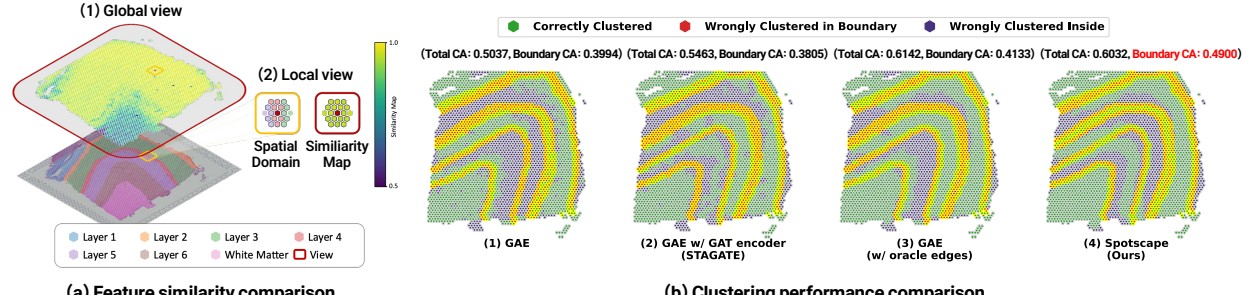

*Figure 1.* (a) Feature similarity comparison from global and local perspectives. In global view, the similarity between the anchor (i.e., red dot) and other spots gradually changes with their spatial coordinates. In contrast, in the local view, neighboring spots exhibit minimal feature discrepancy compared to the anchor, irrespective of the true spatial domain. (b) Clustering performance comparison in terms of clustering accuracy for all spots (Total CA) and particularly for spots located at the boundary of clusters (Boundary CA) in the human dorsolateral prefrontal cortex (DLPFC) dataset.

edge weights[2], and (4) GAE incorporating global similarity learning (our proposed method). While the attention mechanism improves general clustering performance (i.e., Total CA), it degrades the clustering performance of boundary spots (i.e., Boundary CA). This highlights the difficulty of learning spot representations near the boundary of spatial domains using attention. Another interesting observation is that even with oracle edge weights, improvement in terms of boundary CA is not significant compared with the GAE on the original SNN, supporting our argument that solely relying on the local view provides limited information.

In addition to addressing the aforementioned challenges in single-slice analysis, representation learning models for the SRT dataset must account for *batch effects* (Li et al., 2020b) to facilitate multi-slice analysis. The batch effect refers to the phenomenon where gene expression profiles from the same slice cluster together unexpectedly, regardless of their biological relevance, during multi-slice integration. While integrating multiple datasets offers significant advantages, addressing batch effects remains a key challenge.

To this end, we propose Spotscape, a novel framework designed to tackle challenges in both single-slice and multi-slice tasks. Based on our findings that relying solely on spatially local neighbors provides limited performance gains, Spotscape introduces the Similarity Telescope module, which captures relative similarities not only among spatially neighboring spots but also across global spots. This learning scheme is particularly beneficial for SRT data, as optimizing similarity directly supports downstream analyses that involve comparing relative distances, such as clustering or marker gene detection.

Furthermore, we extend Spotscape to multi-slice tasks by employing a prototypical contrastive learning (PCL) scheme to cluster semantically similar spots (i.e., spots with the

___
[2]Edges between spots of the same type were assigned a weight of 1, and 0 otherwise. That is, we remove heterophilic edges.

same domain) from different slices in latent space. Moreover, we propose a similarity scale matching loss that explicitly balances the similarity scales of inter- and intra-relationships to mitigate the batch effect. This strategy enhances the integration of representations across slices, enabling our model to handle both single- and multi-slice SRT data effectively.

In summary, our contributions are four-fold:

- We find that capturing similarity among spatially local neighbors alone is insufficient for learning meaningful representations in SRT data, especially near the boundaries of spatial domains.

- To address this limitation, we propose a global similarity learning scheme called the Similarity Telescope module to capture the relationships between spots in the global context.

- We adopt a PCL scheme and introduce a similarity scale matching strategy to mitigate batch effects during simultaneous training with multiple slices, enabling our model to perform effectively on both single-slice and multi-slice SRT data.

- We conduct extensive experiments across various tasks and datasets to validate the effectiveness of Spotscape in both single- and multi-slice datasets.

## 2. Related Work

### 2.1. Representation learning for SRT data

Learning effective representations of SRT data is critical for downstream tasks, such as spatial domain identification (SDI), which categorizes biologically meaningful tissue regions and enhances our understanding of tissue organization (Maynard et al., 2021). Recently, graph-based deep learning methods have incorporated spatial coordinates and gene expression. For instance, SEDR (Xu et al., 2024a)

employs a graph autoencoder (GAE) with masking to learn and denoise spatial gene expression, while SpaGCN (Hu et al., 2021) integrates spatial and gene expression data using graph neural networks and clustering loss (Xie et al., 2016). STAGATE (Dong & Zhang, 2022) applies graph attention networks to address boundary heterogeneity. Self-supervised learning has also gained popularity for capturing robust representations without labels. SpaceFlow (Ren et al., 2022) utilizes Deep Graph Infomax (DGI) (Veličković et al., 2018) with spatial regularization for spatial consistency, while SpaCAE (Hu et al., 2024) uses a GAE with contrastive learning to handle sparse and noisy SRT data.

## 2.2. Slice Integration and Alignment

Numerous SRT studies collect data from neighboring tissue sections, but inconsistencies in dissection and positioning lead to misaligned spatial coordinates. As a result, integrating data across slices is essential for extracting diverse insights. PASTE (Zeira et al., 2022) addresses this using optimal transport to align spots into a shared embedding space. However, SRT data is sometimes generated under varying conditions (e.g., different technology platforms, developmental stages, or sample conditions). We refer to this as the heterogeneous case, which presents an additional challenge: batch effects, where gene expression profiles from the same slice cluster together, irrespective of their biological significance. STAligner (Zhou et al., 2023) mitigates this by defining mutual nearest neighbors as positive samples and using triplet loss to integrate embeddings across slices. GraphST (Long et al., 2023) employs DGI to correct batch effects by maximizing mutual information in vertical or horizontal integrations. Moreover, PASTE2 (Liu et al., 2023) uses partial optimal transport concept for partial alignment. Furthermore, SLAT (Xia et al., 2023) uses graph adversarial training for robust slice alignment and CAST (Tang et al., 2024) leverages CCA-SSG (Zhang et al., 2021) for heterogeneous slices integration and alignment. Our approach addresses both homogeneous and heterogeneous integration and alignment tasks using a prototypical contrastive learning scheme and simple similarity scale matching strategy.

## 3. Problem Statement

**Notations.** The SRT data is composed of spatial coordinates $S \in \mathbb{R}^{N_s \times 2}$ and gene expression profile $X \in \mathbb{R}^{N_s \times N_g}$, where $N_s$ represents the total number of spots across all slices, and $N_g$ denotes the number of genes. In multi-slice cases, the spatial coordinates and gene expression profiles are denoted as $S = (S^{(1)}, S^{(2)}, \ldots, S^{(N_d)})$ and $X = (X^{(1)}, X^{(2)}, \ldots, X^{(N_d)})$, respectively, where $N_d$ represents the number of slices. We construct spatial nearest neighbors (SNN) graphs $\mathcal{G} = (\mathcal{G}^{(1)}, \mathcal{G}^{(2)}, \ldots, \mathcal{G}^{(N_d)})$ based on distances computed from spatial coordinates and rep-

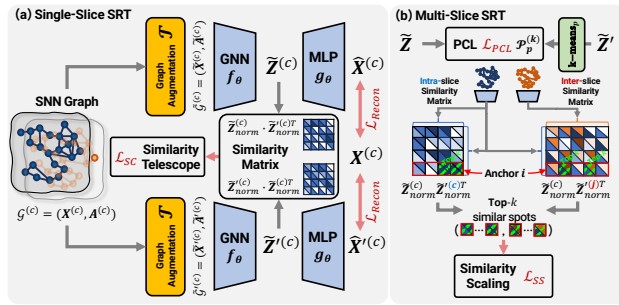

*Figure 2.* Overall framework of Spotscape, which is trained with the SNN graph using (a) similarity telescope and reconstruction loss, while additionally utilizing (b) PCL and similarity scale matching loss in multi-slice SRT.

resent the entire structure as $\mathcal{G} = (X, A)$. The adjacency matrix $A \in \mathbb{R}^{N_s \times N_s}$ is defined such that $A_{ij} = 1$ if there is an edge connecting nodes $i$ and $j$, and $A_{ij} = 0$ otherwise. For simplicity, we do not define a separate adjacency matrix for each slice, resulting in a block-diagonal matrix where all elements are zero across different slices.

**Task description.** Given the constructed SNN graph $\mathcal{G}$, our goal is to train a graph neural network (GNN) that generates spot representations without any label information, i.e., self-supervised learning. The trained GNN is then utilized for various downstream tasks, including SDI, trajectory inference, imputation, multi-slice integration, and alignment.

## 4. Methodology

In this section, we introduce our method, Spotscape. In a nutshell, Spotscape learns spot representations by capturing global similarities between spots through the Similarity Telescope module (Sec 4.2) for the single-slice tasks. Moreover, to enhance multi-slice tasks, we adopt a prototypical contrastive learning module (Sec 4.3) to group spots from the same spatial domain across different slices and introduce a similarity scaling strategy (Sec 4.4) to balance intra- and inter-slice similarities, thereby alleviating batch effects. The overall framework of Spotscape is depicted in Figure 2.

### 4.1. Model Architecture

In this work, we propose novel self-supervised learning strategies specifically tailored for SRT data, while adhering to a basic siamese network structure for our model architecture. In siamese network, we generate two augmented views, $\tilde{\mathcal{G}} = (\tilde{X}, \tilde{A})$ and $\tilde{\mathcal{G}}' = (\tilde{X}', \tilde{A}')$, by applying a stochastic graph augmentation $\mathcal{T}$ to the original graph $\mathcal{G}$, which consists of node feature masking and edge masking. Then, Spotscape computes spot representations $\tilde{Z} = f_\theta(\tilde{X}, \tilde{A})$ and $\tilde{Z}' = f_\theta(\tilde{X}', \tilde{A}')$, $f_\theta$ is a shared GNN-based encoder, $\tilde{Z} \in \mathbb{R}^{N_s \times D}$ and $\tilde{Z}' \in \mathbb{R}^{N_s \times D}$ represent spot representations derived from augmented graph $\tilde{\mathcal{G}}$ and $\tilde{\mathcal{G}}'$, respectively,

and $D$ denotes the dimension size of representations.

## 4.2. Similarity Telescope with Relation Consistency

Biological systems exhibit a continuous nature, where gene expression values vary smoothly along spatial coordinates. This continuity results in feature similarities between neighboring spots, influenced by both meaningful functional characteristics and spatial proximity. Consequently, spatially neighboring spots provide limited information, as their feature differences are not fully obtained from meaningful ones and are often merely close due to their spatial proximity. Therefore, relying solely on spatially neighboring spots is insufficient for accurate representation learning, highlighting the importance of reflecting the global context in this domain. To this end, we propose a novel relation consistency loss for spot representation learning, which aims to capture the relationship between cells in the biological systems by reflecting the global context among multiple spots.

Specifically, given spot representations $\tilde{Z}$ and $\tilde{Z}'$, we propose to learn the consistent relationship that are invariant under augmentation as follows:

$$\mathcal{L}_{\text{SC}}(\tilde{Z}, \tilde{Z}') = \text{MSE}(\tilde{Z}_{\text{norm}}(\tilde{Z}'_{\text{norm}})^T, \tilde{Z}'_{\text{norm}}(\tilde{Z}_{\text{norm}})^T) \quad (1)$$

where $\tilde{Z}_{\text{norm}} \in \mathbb{R}^{N_s \times D}$ denotes the L2-normalized version of $\tilde{Z}$, and $\text{MSE}$ represents the Mean Squared Error. That is, we aim to minimize the cosine similarity between the spot representations that are obtained through differently augmented SNN graph. By doing so, the model learns consistent relationships, which is represented as cosine similarity, between all paired spots under different augmentations, capturing the continuous variations of spot representations across the entire slice. A more detailed interpretation of the relation consistency loss $\mathcal{L}_{\text{SC}}$ can be found in Appendix L.

To avoid degenerate solutions, Spotscape employs a reconstruction loss as follows:

$$\mathcal{L}_{\text{Recon}}(X, \hat{X}, \hat{X}') = \text{MSE}(X, \hat{X}) + \text{MSE}(X, \hat{X}') \quad (2)$$

where $\hat{X} = g_\theta(\tilde{Z})$ and $\hat{X}' = g_\theta(\tilde{Z}')$ are reconstructed feature matrices predicted by a shared MLP decoder $g_\theta$ from each augmented view. Note that utilizing this reconstruction module offers additional advantages, as the reconstructed output (i.e., imputed data) can be valuable for imputing and denoising raw transcriptomics data.

Combining all these two losses, the final training loss for single-slice representation learning is formally defined as:

$$\mathcal{L}_{\text{Single}} = \lambda_{\text{SC}} \mathcal{L}_{\text{SC}} + \lambda_{\text{Recon}} \mathcal{L}_{\text{Recon}} \quad (3)$$

## 4.3. Prototypical Contrastive Learning

Beyond single-slice SRT, multi-slice SRT analysis elucidates the spatial regulatory mechanisms of specific biolog-

ical processes by analyzing the continuity of gene expression across the entire tissue or organ. For this analysis, researchers need to identify corresponding or similar spots across different slices, requiring an integrated representation space. To ensure a well-integrated representation space, spots should be merged based on meaningful characteristics. To this end, Spotscape employs a prototypical contrastive learning (PCL) scheme (Li et al., 2020a; De Donno et al., 2023; Lee et al., 2023) to group spots with the same spatial domain while distancing others in latent space. Specifically, we obtain prototypes (i.e., centroids) by performing $K$-means clustering on spot representations $\tilde{Z}'$ derived from an augmented view $\tilde{\mathcal{G}}'$. Pairs of spots assigned to the same prototype are categorized as positive pairs, while pairs belonging to different prototypes are treated as negative pairs. This clustering is repeated $T$ times with varying values of $K$ to identify semantically similar groups across different granularities. It is formally represented as follows:

$$l_{\text{PCL}}(\tilde{Z}_i, P_{\text{set}}) = \frac{1}{T} \sum_{t=1}^{T} \log \frac{e^{(\text{sim}(\tilde{Z}_i, p^t_{\text{map}_t(i)})/\tau)}}{\sum_{j=1}^{K_t} e^{(\text{sim}(\tilde{Z}_i, p^t_j)/\tau)}}, \quad (4)$$

where $\tau$ represents temperature, and $K_t$ indicates the number of clusters at each level of granularity during the $t$-th clustering iteration. $P_{\text{set}} = (P^1, ..., P^t, ..., P^T)$ represents the collection of prototype sets, with each $P^t = (p^t_1, p^t_2, ..., p^t_{k_t})$ containing the set of prototype representations for a specific granularity $t$. Additionally, $\text{map}_t(\cdot)$ denotes the mapping function that assigns each spot to a corresponding prototype based on the clustering assignments. By applying this to all spot representations, the overall PCL loss is given as follows:

$$\mathcal{L}_{\text{PCL}} = -\frac{1}{N_s} \sum_{i=1}^{N_s} l_{\text{PCL}}(\tilde{Z}_i, P_{\text{set}}). \quad (5)$$

Note that to avoid the risk of obtaining inaccurate prototypes, the PCL loss $\mathcal{L}_{\text{PCL}}$ gets involved in the training procedure after a warm-up period (500 epochs). This loss could be applied to single-slice cases; however, since the representations in this case are better grouped than in multi-slice cases, we chose not to use it due to the trade-off with running time. We discuss this trade-off in Appendix F.

## 4.4. Similarity Scaling Strategy

The primary challenge of learning representations from multiple slices is the batch effect, which causes representations from the same slice to cluster together unexpectedly, regardless of their biological significance. From a computational perspective, this means that a given spot's top-$k$ nearest neighbors from the same batch exhibit higher similarity than those from different batches, causing its representation space to be dominated by same-batch spots.

To alleviate this issue, given the SNN graph $\mathcal{G}^{(c)}$ and $\mathcal{G}^{(j)}$ of the current slice $c$ and another slice $j$, respectively, we

explicitly regulate the scale of these similarities to maintain consistency across spots, as described below:

$$l_{\text{SS}}(H_i, \mathcal{G}^{(j)}) = (\text{Mean}(S_{\text{top}}^{(c)}) - \text{Mean}(S_{\text{top}}^{(j)}))^2, \text{for } i \in \mathcal{G}^{(c)}$$

$$\text{where} \quad S_{\text{top}}^{(c)} = \text{Top-}k_{l \in \mathcal{G}^{(c)}}(H_i[l]) = (a_1, a_2, \ldots, a_k),$$

$$S_{\text{top}}^{(j)} = \text{Top-}k_{l \in \mathcal{G}^{(j)}}(H_i[l]) = (b_1, b_2, \ldots, b_k)$$

(6)

Here, $H = \tilde{Z}_{\text{norm}}(\tilde{Z}'_{\text{norm}})^T \in \mathbb{R}^{N_s \times N_s}$ represents the similarity matrix that we optimize in the Similarity Telescope module, and $H_i[s]$ refers to the element in the $i$-th row and $s$-th column of this matrix. $S_{\text{top}}^{(c)}$ is the list of the top-$k$ highest similarity values for spot $i$ within its own slice, and the list $S_{\text{top}}^{(j)}$ consists of the top-$k$ highest similarity values for spot $i$ in another slice $j$. By doing this, Spotscape ensures that the distances between the top-$k$ spots remain nearly the same, regardless of their slice, effectively incorporating all spots from different slices within the latent space. By extending it to all spots and slices, the final similarity scaling loss is given as follows:

$$\mathcal{L}_{\text{SS}} = \frac{1}{N_s(N_d - 1)} \sum_{i=1}^{N_s} \sum_{j=1}^{N_d} \mathbb{1}(i \notin \mathcal{G}^{(j)}) \cdot l_{\text{SS}}(H_i, \mathcal{G}^{(j)})$$

(7)

where $\mathbb{1}(i \notin \mathcal{G}^{(j)})$ is the indicator function that equals 1 if spot $i$ is not included in $\mathcal{G}^{(j)}$ and 0 otherwise. Note that since this loss matches similarities across different slices, it is not applicable to the single-slice case. Finally, the overall loss for multi-slice SRT data is formally represented as:

$$\mathcal{L}_{\text{Multi}} = \lambda_{\text{SC}}\mathcal{L}_{\text{SC}} + \lambda_{\text{Recon}}\mathcal{L}_{\text{Recon}} + \lambda_{\text{PCL}}\mathcal{L}_{\text{PCL}} + \lambda_{\text{SS}}\mathcal{L}_{\text{SS}} \quad (8)$$

where $\lambda_{\text{PCL}}$, $\lambda_{\text{SS}}$ are additional balancing parameters of prototypical contrastive learning loss and similarity scaling loss, respectively. The pseudo-code for Spotscape is provided in Appendix C.

## 5. Experiments

**Datasets.** We conduct a comprehensive evaluation of Spotscape across five datasets derived from different sequencing technologies. For **single-slice experiments**, we use the dorsolateral prefrontal cortex (**DLPFC**) dataset, which includes 3 patients, each with 4 slices (12 slices in total). Additionally, we assess the middle temporal gyrus (**MTG**) dataset, comprising slices from a control group and an Alzheimer's disease (AD) group, as well as the **Mouse embryo** dataset. Lastly, we utilize non-small cell lung cancer (**NSCLC**) data obtained from CosMX sequencing, which provides one of the highest subcellular resolutions among sequencing platforms. In **multi-slice experiments**, we integrate the four slices from the same patient in the **DLPFC** dataset for the homogeneous integration task, while analyzing the differences between the control and

AD groups in the **MTG** dataset for heterogeneous integration. Lastly, we evaluate heterogeneous alignment using the **Mouse embryo** dataset, where slices from different developmental stages require alignment to track developmental progression, and the **Breast Cancer** dataset, which includes spots corresponding to cancer cell types. Further details about data statistics can be found in Table 5 of Appendix A.

**Compared methods.** To ensure a fair comparison, we carefully select baseline methods based on their relevance to specific tasks. For the single-slice tasks, we compare Spotscape with five state-of-the arts methods, i.e., SEDR (Xu et al., 2024a), STAGATE (Dong & Zhang, 2022), SpaCAE (Hu et al., 2024), SpaceFlow (Ren et al., 2022), and GraphST (Long et al., 2023). For homogeneous integration, we add three more methods, PASTE (Zeira et al., 2022), STAligner (Zhou et al., 2023), and CAST (Tang et al., 2024). For heterogeneous tasks, we compare with GraphST, STAligner (Zhou et al., 2023), and CAST, while for heterogeneous alignment, we compare with PASTE2 (Liu et al., 2023), CAST, STAligner, and SLAT (Xia et al., 2023), both specialized for alignment tasks. Further details about each method's adoptable application can be found in Table 6 of Appendix B.

**Evaluation protocol.** Since Spotscape and all baseline methods focus on learning spot representations, we first obtain representations from each method and apply the same evaluation tools for downstream tasks. For single-slice spatial domain identification, we apply $K$-means clustering and evaluate performance using Adjusted Rand Index (ARI), Normalized Mutual Information (NMI), and Clustering Accuracy (CA). For trajectory inference, we compute the pseudo-dotime following Spaceflow (Ren et al., 2022) and measure its spearman correlation with gold-standard layers with scaling. For multi-slice integration, we assess clustering using the same metrics as in the single-slice experiments and additionally evaluate batch effect correction using Silhouette Batch, kBET, Graph Connectivity, and PCR comparison. For alignment, we utilize the 'spatial matching' function from SLAT (Xia et al., 2023) and evaluate the alignment quality using Label Transfer ARI (LTARI), which measures the agreement between true and the aligned labels. To ensure fairness, we conduct a hyperparameter search for all baseline methods instead of using their default settings, as optimal hyperparameters may vary across datasets. The best-performing hyperparameters are determined based on NMI using the first seed. For Spotscape, only the learning rate is searched using the same criterion. Details of the selected hyperparameters and search spaces are provided in Appendix E.1, along with an unsupervised approach for hyperparameter selection in Appendix E.2 to address potential concerns about our evaluation strategy. All experiments are repeated over 10 runs with different random seeds, and we report the mean and standard deviation of the results. For all

*Table 1.* Single-slice spatial domain identification performance on (a) DLPFC, (b) MTG, (c) Mouse Embryo, and (d) NSCLC datasets.

**(a) DLPFC (Patient 1)**

| | Slice 151673 | | | Slice 151674 | | | Slice 151675 | | | Slice 151676 | | |
|---|---|---|---|---|---|---|---|---|---|---|---|---|
| | ARI | NMI | CA | ARI | NMI | CA | ARI | NMI | CA | ARI | NMI | CA |
| SEDR | 0.36 (0.08) | 0.49 (0.08) | 0.55 (0.06) | 0.37 (0.08) | 0.48 (0.07) | 0.51 (0.07) | 0.33 (0.06) | 0.45 (0.05) | 0.51 (0.05) | 0.29 (0.03) | 0.41 (0.04) | 0.47 (0.02) |
| STAGATE | 0.37 (0.04) | 0.55 (0.03) | 0.52 (0.04) | 0.34 (0.03) | 0.50 (0.02) | 0.51 (0.03) | 0.33 (0.03) | 0.50 (0.01) | 0.48 (0.03) | 0.33 (0.00) | 0.47 (0.01) | 0.52 (0.01) |
| SpaCAE | 0.21 (0.01) | 0.37 (0.01) | 0.43 (0.01) | 0.25 (0.03) | 0.38 (0.01) | 0.44 (0.03) | 0.23 (0.03) | 0.41 (0.03) | 0.42 (0.04) | 0.23 (0.02) | 0.34 (0.02) | 0.43 (0.03) |
| SpaceFlow | 0.42 (0.06) | 0.57 (0.05) | 0.57 (0.03) | 0.37 (0.04) | 0.51 (0.03) | 0.53 (0.03) | 0.38 (0.07) | 0.55 (0.06) | 0.53 (0.05) | 0.38 (0.05) | 0.51 (0.05) | 0.53 (0.04) |
| GraphST | 0.20 (0.02) | 0.34 (0.03) | 0.41 (0.02) | 0.27 (0.02) | 0.41 (0.01) | 0.46 (0.01) | 0.22 (0.02) | 0.34 (0.01) | 0.40 (0.02) | 0.26 (0.05) | 0.40 (0.05) | 0.45 (0.04) |
| Spotscape | **0.48** ** (0.02) | **0.64** ** (0.01) | **0.61** ** (0.02) | **0.47** * (0.04) | **0.60** ** (0.02) | **0.60** ** (0.02) | **0.45** * (0.02) | **0.60** * (0.01) | **0.59** * (0.02) | **0.42** * (0.05) | **0.58** ** (0.04) | **0.57** * (0.03) |

**(a) DLPFC (Patient 2)**

| | Slice 151507 | | | Slice 151508 | | | Slice 151509 | | | Slice 151510 | | |
|---|---|---|---|---|---|---|---|---|---|---|---|---|
| | ARI | NMI | CA | ARI | NMI | CA | ARI | NMI | CA | ARI | NMI | CA |
| SEDR | 0.29 (0.06) | 0.39 (0.07) | 0.45 (0.06) | 0.21 (0.02) | 0.31 (0.02) | 0.47 (0.04) | 0.37 (0.04) | 0.47 (0.04) | 0.51 (0.05) | 0.31 (0.03) | 0.44 (0.04) | 0.47 (0.04) |
| STAGATE | 0.41 (0.01) | 0.53 (0.01) | 0.59 (0.00) | 0.32 (0.01) | 0.49 (0.00) | 0.54 (0.01) | 0.41 (0.01) | 0.57 (0.02) | 0.61 (0.04) | 0.32 (0.03) | 0.50 (0.02) | 0.50 (0.02) |
| SpaCAE | 0.28 (0.06) | 0.41 (0.06) | 0.46 (0.06) | 0.20 (0.04) | 0.31 (0.05) | 0.40 (0.04) | 0.31 (0.01) | 0.44 (0.02) | 0.50 (0.04) | 0.27 (0.02) | 0.42 (0.03) | 0.45 (0.02) |
| SpaceFlow | 0.55 (0.03) | 0.68 (0.02) | 0.71 (0.05) | 0.44 (0.04) | 0.57 (0.03) | 0.58 (0.04) | 0.53 (0.05) | 0.66 (0.02) | 0.65 (0.04) | 0.50 (0.04) | 0.64 (0.02) | 0.61 (0.02) |
| GraphST | 0.31 (0.01) | 0.45 (0.01) | 0.50 (0.01) | 0.34 (0.01) | 0.45 (0.02) | 0.53 (0.02) | 0.35 (0.01) | 0.51 (0.01) | 0.55 (0.02) | 0.30 (0.02) | 0.47 (0.01) | 0.49 (0.03) |
| Spotscape | **0.60** ** (0.03) | **0.72** ** (0.01) | **0.76** ** (0.03) | **0.48** * (0.05) | **0.64** ** (0.03) | **0.63** ** (0.02) | **0.59** ** (0.01) | **0.71** ** (0.01) | **0.70** ** (0.02) | **0.53** * (0.04) | **0.67** ** (0.02) | **0.64** (0.04) |

**(a) DLPFC (Patient 3)**

| | Slice 151669 | | | Slice 151670 | | | Slice 151671 | | | Slice 151672 | | |
|---|---|---|---|---|---|---|---|---|---|---|---|---|
| | ARI | NMI | CA | ARI | NMI | CA | ARI | NMI | CA | ARI | NMI | CA |
| SEDR | 0.24 (0.07) | 0.40 (0.07) | 0.48 (0.06) | 0.24 (0.06) | 0.40 (0.05) | 0.48 (0.05) | 0.37 (0.04) | 0.50 (0.04) | 0.59 (0.07) | 0.49 (0.09) | 0.58 (0.06) | 0.66 (0.07) |
| STAGATE | 0.29 (0.05) | 0.45 (0.07) | 0.52 (0.04) | 0.20 (0.01) | 0.38 (0.01) | 0.44 (0.01) | 0.40 (0.07) | 0.49 (0.03) | 0.63 (0.06) | 0.38 (0.02) | 0.51 (0.04) | 0.54 (0.01) |
| SpaCAE | 0.21 (0.02) | 0.28 (0.03) | 0.43 (0.02) | 0.21 (0.03) | 0.28 (0.02) | 0.43 (0.04) | 0.38 (0.16) | 0.29 (0.01) | 0.49 (0.05) | 0.25 (0.04) | 0.35 (0.05) | 0.50 (0.01) |
| SpaceFlow | 0.30 (0.07) | 0.48 (0.03) | 0.51 (0.05) | 0.34 (0.03) | 0.50 (0.03) | 0.56 (0.04) | 0.54 (0.04) | 0.67 (0.02) | 0.67 (0.04) | 0.60 (0.04) | 0.70 (0.02) | 0.73 (0.06) |
| GraphST | 0.17 (0.04) | 0.26 (0.04) | 0.43 (0.02) | 0.14 (0.01) | 0.23 (0.00) | 0.37 (0.01) | 0.30 (0.05) | 0.38 (0.03) | 0.54 (0.03) | 0.23 (0.01) | 0.32 (0.02) | 0.49 (0.02) |
| Spotscape | **0.46** ** (0.02) | **0.58** ** (0.02) | **0.65** ** (0.02) | **0.45** ** (0.04) | **0.56** ** (0.02) | **0.66** ** (0.03) | **0.68** ** (0.10) | **0.74** ** (0.02) | **0.79** ** (0.08) | **0.75** ** (0.04) | **0.74** ** (0.02) | **0.84** ** (0.05) |

| | (b) MTG - Control Group | | | (b) MTG - AD Group | | | (c) Mouse Embryo | | | | (d) NSCLC | | |
|---|---|---|---|---|---|---|---|---|---|---|---|---|---|---|
| | ARI | NMI | CA | ARI | NMI | CA | | ARI | NMI | CA | | ARI | NMI | CA |
| SEDR | 0.41 (0.02) | 0.59 (0.02) | 0.52 (0.02) | 0.43 (0.06) | 0.59 (0.07) | 0.57 (0.07) | SEDR | 0.32 (0.02) | 0.56 (0.01) | 0.42 (0.02) | SEDR | 0.44 (0.06) | 0.46 (0.04) | 0.70 (0.06) |
| STAGATE | 0.54 (0.00) | 0.65 (0.00) | 0.59 (0.00) | 0.51 (0.01) | 0.61 (0.01) | 0.59 (0.01) | STAGATE | 0.36 (0.01) | 0.60 (0.01) | 0.47 (0.01) | STAGATE | 0.35 (0.05) | 0.41 (0.04) | 0.64 (0.02) |
| SpaCAE | 0.37 (0.03) | 0.52 (0.00) | 0.44 (0.03) | 0.22 (0.01) | 0.40 (0.01) | 0.40 (0.01) | SpaCAE | 0.34 (0.01) | 0.60 (0.01) | 0.48 (0.02) | SpaCAE | 0.32 (0.05) | 0.38 (0.03) | 0.62 (0.02) |
| SpaceFlow | 0.66 (0.03) | 0.74 (0.01) | 0.70 (0.03) | 0.54 (0.01) | 0.71 (0.00) | 0.65 (0.01) | SpaceFlow | 0.42 (0.03) | 0.60 (0.02) | 0.49 (0.03) | SpaceFlow | 0.53 (0.03) | 0.52 (0.02) | 0.75 (0.02) |
| GraphST | 0.38 (0.00) | 0.51 (0.00) | 0.48 (0.00) | 0.43 (0.06) | 0.55 (0.05) | 0.55 (0.04) | GraphST | 0.34 (0.01) | 0.59 (0.02) | 0.45 (0.01) | GraphST | 0.30 (0.00) | 0.38 (0.00) | 0.65 (0.00) |
| Spotscape | **0.73** ** (0.02) | **0.78** ** (0.01) | **0.75** ** (0.03) | **0.68** ** (0.02) | **0.75** ** (0.01) | **0.77** ** (0.03) | Spotscape | **0.44** (0.01) | **0.63** ** (0.01) | **0.54** ** (0.01) | Spotscape | **0.57** ** (0.02) | **0.57** ** (0.01) | **0.74** (0.01) |

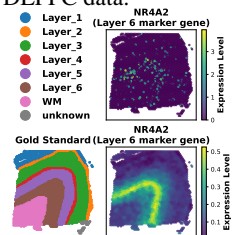

*Figure 3.* Trajectory conservation between pseudotimes and Layers in DLPFC data.

*Figure 4.* Spatial expression of raw and Spotscape imputed data for marker genes of Layer 6 in DLPFC data.

experimental results, **Bold** indicates the best performance, underlining denotes the second-best, and an asterisk (*) marks statistically significant improvements of Spotscape over the top-performing baseline based on a paired t-test (**: $p < 0.01$, *: $p < 0.05$), with the numbers in parentheses representing the standard deviation.

## 5.1. Single-slice Experimental Results

**Spatial domain identification (SDI).** Experimental results on four different datasets are reported in Table 1, which shows the SDI performance on (a) DLPFC, (b) MTG, (c) Mouse Embryo, and (d) NSCLC datasets, respectively. From these results, we have the following observations: **1)** Spotscape consistently outperforms in all 16 slices across four datasets in terms of ARI, NMI, and CA. We argue that this is because Spotscape not only explores information from spatially local neighbors, which provides limited insights due to the continuous nature of SRT data, but also leverages global contextual information. **2)** SpaceFlow also shows high performance compared to other baselines by mapping the spatial distance between spots to the representation space through regularization loss. This suggests that accurately capturing the distance between spots is crucial for SRT data analysis. **3)** However, it still exhibits lower performance compared to Spotscape, as the proposed similarity telescope module directly optimizes the similarities (i.e., distances) between spots by maintaining consistency between augmentations, thereby providing more accurate similarities. Further analysis of this learned similarity is

provided in Section 5.3, and a comparison to general self-supervised learning methods can be found in Appendix B.

**Trajectory inference.** To further validate the broad applicability of Spotscape, we conducted trajectory inference on DLPFC data. The results, shown in Figure 3, demonstrate that Spotscape effectively performs trajectory inference and accurately captures biologically meaningful spatiotemporal patterns. Specifically, Spotscape reveals a layer-patterned pseudotime, indicating a pseudo-spatiotemporal order from White Matter to Layer 1, which aligns with the correct inside-out developmental sequence of cortical layers and reflects the tissue's layered spatial organization. Additional details can be found in Appendix H.

**Imputation.** To demonstrate the benefits of incorporating a decoder layer and reconstruction loss, we perform imputation tasks to show that the reconstructed outputs can help identify marker genes that were not differentially expressed in the raw data. In Figure 4, we show that NR4A2, a marker for layer 6 neurons (Maynard et al., 2021; Darbandi et al., 2018), which was not well-recognized in the raw data, becomes more distinct in the denoised output. Comprehensive results on additional marker gene detection and quantitative comparisons with baselines are provided in Appendix I.

## 5.2. Multi-slice Experimental Results

**Homogeneous integration.** Among the multi-slice experiments, we first start with homogeneous integration tasks, which aim to integrate multiple slices from the homoge-

*Table 2.* Homogeneous integration performance on DLPFC data.

| | Patient 1 | | | Patient 2 | | | Patient 3 | | |
|---|---|---|---|---|---|---|---|---|---|
| | **ARI** | **NMI** | **CA** | **ARI** | **NMI** | **CA** | **ARI** | **NMI** | **CA** |
| SEDR | 0.38 (0.06) | 0.49 (0.06) | 0.56 (0.06) | 0.32 (0.05) | 0.44 (0.07) | 0.48 (0.07) | 0.43 (0.02) | 0.51 (0.01) | 0.56 (0.03) |
| STAGATE | 0.31 (0.03) | 0.46 (0.03) | 0.49 (0.03) | 0.30 (0.02) | 0.46 (0.01) | 0.48 (0.02) | 0.31 (0.09) | 0.43 (0.06) | 0.54 (0.08) |
| SpaCAE | 0.21 (0.03) | 0.36 (0.02) | 0.40 (0.02) | 0.12 (0.06) | 0.19 (0.07) | 0.32 (0.05) | 0.13 (0.05) | 0.14 (0.05) | 0.43 (0.06) |
| SpaceFlow | 0.48 (0.03) | 0.60 (0.02) | 0.60 (0.02) | 0.44 (0.05) | 0.59 (0.02) | 0.58 (0.04) | 0.51 (0.02) | 0.60 (0.01) | 0.69 (0.05) |
| GraphST | 0.18 (0.01) | 0.32 (0.01) | 0.38 (0.02) | 0.25 (0.01) | 0.39 (0.01) | 0.42 (0.02) | 0.25 (0.04) | 0.30 (0.04) | 0.50 (0.01) |
| PASTE | 0.34 (0.00) | 0.45 (0.00) | 0.54 (0.00) | 0.17 (0.00) | 0.28 (0.00) | 0.40 (0.00) | 0.29 (0.00) | 0.43 (0.00) | 0.54 (0.00) |
| STAligner | 0.38 (0.04) | 0.52 (0.04) | 0.55 (0.04) | 0.29 (0.02) | 0.45 (0.02) | 0.48 (0.03) | 0.37 (0.06) | 0.47 (0.05) | 0.59 (0.06) |
| CAST | 0.26 (0.02) | 0.37 (0.03) | 0.42 (0.03) | 0.30 (0.04) | 0.43 (0.05) | 0.47 (0.03) | 0.38 (0.06) | 0.40 (0.04) | 0.56 (0.05) |
| Spotscape | **0.57** (0.03) | **0.70** (0.02) | **0.67** (0.03) | **0.53** (0.02) | **0.67** (0.01) | **0.63** (0.02) | **0.63** (0.09) | **0.68** (0.03) | **0.75** (0.09) |

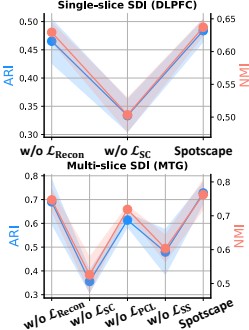

*Figure 5.* Alignment results of Mouse embryo data.

*Table 3.* Heterogeneous integration performance on MTG data.

| | Clustering Metric | | | Batch Effect Correction Metric | | | |
|---|---|---|---|---|---|---|---|
| | **ARI** | **NMI** | **CA** | **Silhouette batch** | **kBET** | **Graph connectivity** | **PCR comparison** |
| GraphST | 0.23 (0.02) | 0.42 (0.00) | 0.39 (0.01) | 0.56 (0.00) | 0.02 (0.01) | 0.65 (0.02) | 0.00 (0.00) |
| STAligner | 0.38 (0.03) | 0.54 (0.03) | 0.49 (0.02) | 0.62 (0.04) | 0.11 (0.08) | 0.85 (0.04) | 0.18 (0.10) |
| CAST | 0.48 (0.07) | 0.52 (0.06) | 0.59 (0.06) | 0.45 (0.02) | **0.11** (0.02) | 0.81 (0.06) | **0.97** (0.03) |
| Spotscape (w/o $\mathcal{L}_{PCL}$) | 0.61 (0.03) | 0.71 (0.01) | 0.70 (0.02) | 0.67 (0.01) | 0.03 (0.00) | 0.79 (0.03) | 0.50 (0.04) |
| Spotscape (w/o $\mathcal{L}_{SS}$) | 0.47 (0.09) | 0.60 (0.04) | 0.59 (0.06) | 0.24 (0.01) | 0.00 (0.00) | 0.63 (0.00) | 0.00 (0.00) |
| Spotscape | **0.72** (0.04) | **0.76** (0.01) | **0.81** (0.05) | **0.69** (0.01) | 0.08 (0.02) | **0.86** (0.03) | 0.60 (0.08) |

*Table 4.* Alignment performance of Mouse embryo data.

| | **LTARI** |
|---|---|
| PASTE2 | 0.21 (0.02) |
| CAST | 0.10 (0.01) |
| STAligner | 0.46 (0.01) |
| SLAT | 0.41 (0.11) |
| Spotscape | **0.51** (0.01) |

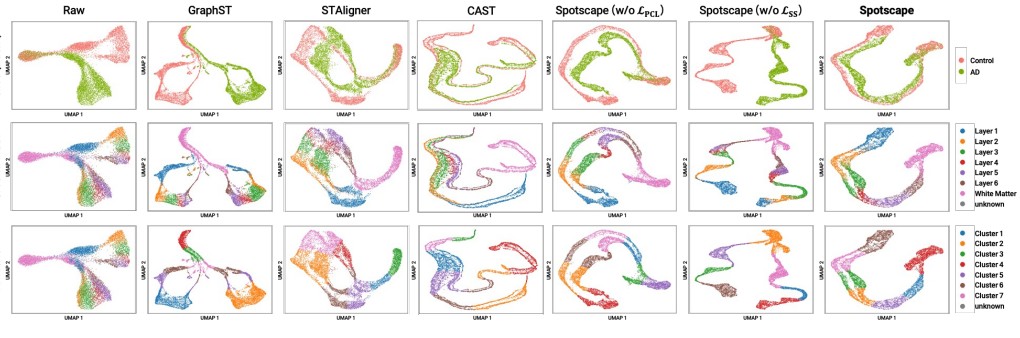

*Figure 6.* UMAP of Raw, GraphST, STAligner, CAST, Spotscape (w/o $\mathcal{L}_{SS}$), Spotscape (w/o $\mathcal{L}_{PCL}$), Spotscape by slice, ground truth, and $K$-means clustering results.

*Figure 7.* Ablation studies.

neous sample. To do so, we conduct experiments on the DLPFC data, which consists of multiple slices obtained from vertical cuts of a single patient. Since these slices are from a single patient, they do not exhibit significant batch effects, enabling us to incorporate both multi-slice integration methods as well as single-slice SDI methods as baselines. As shown in Table 2, we observe that Spotscape consistently outperforms all baseline methods, demonstrating its effectiveness in integrating spots from the multiple slices.

**Heterogeneous integration.** For the heterogeneous integration experiments, we assess the model's ability in integrating two distinct types of samples—the control (CT) group and the Alzheimer's disease (AD) group in the MTG data—to analyze the differences between them. In this experiment, we also report batch effect correction metrics to evaluate the effectiveness of correcting batch effects, along with clustering metrics. In Table 3, Spotscape demonstrates its effectiveness in integrating multi-slice data in terms of both clustering and batch effect correction, showing significantly better performance than the baselines. Moreover, in Figure 6, we observe that Spotscape's spot representations

from different slices are well integrated while preserving their biological meaning. We also verified that both PCL and similarity scaling perform as intended by evaluating their effects after removing each component. Without PCL (i.e., Spotscape w/o $\mathcal{L}_{PCL}$), we observe a drop in clustering performance, as the spot representations are not tightly condensed in the representation space. Additionally, we observe that the batch effect becomes severe without the similarity scaling module (i.e., Spotscape w/o $\mathcal{L}_{SS}$), resulting in a significant degradation in clustering performance, which highlights the importance of this module to alleviate the batch effect for handling multiple slices.

**Differentially expressed gene analysis.** We verify that our results yield biologically meaningful insights by investigating differentially expressed genes (DEGs) and their biological functions between the Control and AD groups using Gene Ontology (GO) enrichment analysis for each cluster representing a cortical layer. Spotscape reveals that layer 2 reflects early-stage AD processes (e.g., oxidative stress), while layer 5 captures later-stage events (e.g., inclusion body assembly), aligning with established AD

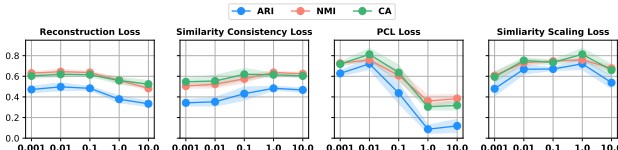

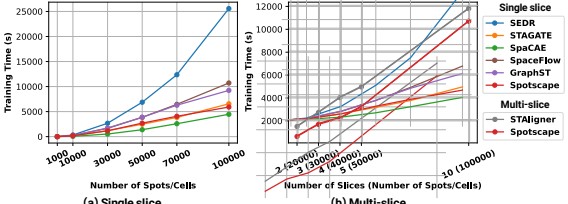

*Figure 8.* Sensitivity analysis of loss balancing parameters.

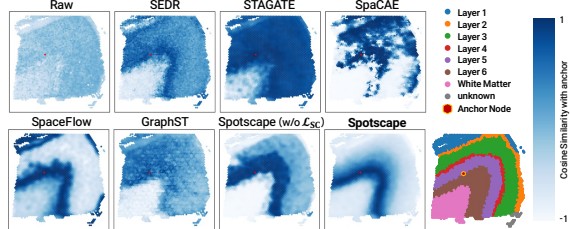

*Figure 10.* The running time of Spotscape and baseline methods over the various number of spots on (a) the single and (b) multi-slice dataset.

*Figure 9.* Similarity comparison based on anchor node 489 in Layer 5.

scaling ($\mathcal{L}_{SS}$) demonstrates its necessity, with a significant performance drop observed without this module. Additionally, we perform ablation studies on encoder variations in the Appendix F.

pathology and showcasing Spotscape's utility in uncovering meaningful biological variation. The detailed results are provided in Appendix J due to space limitations.

**Sensitivity analysis.** We conduct a sensitivity analysis on all four balancing parameters $\lambda_{Recon}$, $\lambda_{SC}$, $\lambda_{PCL}$, and $\lambda_{SS}$ in Figure 8. The reconstruction loss and similarity consistency loss demonstrate robustness across a wide range of values. However, when the weight of the reconstruction loss ($\lambda_{Recon}$) is excessively high, performance tends to degrade, indicating that it serves primarily as an auxiliary loss to prevent degenerate solutions. In contrast, the relation consistency loss ($\lambda_{SC}$) leads to performance degradation when its weight is too small, highlighting its critical role in capturing global similarities within Spotscape. The prototypical contrastive loss ($\lambda_{PCL}$) negatively impacts performance when assigned excessive weight. We attribute this to its tendency to group spots from different domains in the latent space when it dominates the overall training process. Finally, the similarity scaling ($\lambda_{SS}$) maintains robust performance, except when set to extremely low or high values.

**Multi-slice alignment.** We conduct experiments on multi-slice alignments of the Mouse Embryo data, which require alignment results to track the development stages of the embryo. To this end, we match E11.5 and E12.5 and report the Label Transfer ARI (LTARI) in Table 4, which measures the agreement between true labels and the labels assigned through the alignment process, and visualize our results in Figure 5. These results show that Spotscape achieves better alignment than SLAT, which is specifically designed for alignment tasks, demonstrating the general applicability of Spotscape. Furthermore, we conduct cross-technology alignment between data obtained from Xenium and Visium, as reported in Appendix K, to demonstrate that Spotscape can successfully align spots from more heterogeneous scenarios.

**Similarity analysis.** As a deeper analysis of Spotscape, we examine whether it successfully learns the relative similarities between spots, which is a key motivation behind our approach. In Figure 9, we randomly select an anchor spot from the DLPFC data and visualize the similarity between the selected anchor and other remaining spots. While other baselines fail to capture appropriate similarities, Spotscape accurately reflects the dynamics of the SRT data with respect to the spatial distance and exhibits varying levels of similarity corresponding to true spatial domain.

### 5.3. Model Analysis

**Ablation studies.** We conduct ablation studies on the components of Spotscape to clarify the necessity of each module, as shown in Figure 7. Across both tasks, our proposed Similarity Telescope (i.e., $\mathcal{L}_{SC}$) demonstrates its importance by showing a significant performance drop without this module, highlighting the value of incorporating global context. In contrast, the reconstruction loss ($\mathcal{L}_{Recon}$) does not show significant performance gains, as it mainly serves to prevent degenerate solutions. In these experiments, results show that degeneracy does not occur without this module, but it is necessary to address potential issues when applying to other datasets. Additionally, prototypical contrastive learning (i.e., $\mathcal{L}_{PCL}$) further confirms its role in grouping semantically similar spots in latent space by consistently showing performance improvements. Similarly, similarity

**Scalability.** With recent advancements in high-throughput sequencing technologies, model scalability has become a crucial factor in performance evaluation. To assess this, we generate a synthesized dataset by downsampling or oversampling the Mouse Embryo dataset, creating datasets ranging from 1,000 to 100,000 spots. We report the corresponding runtime in Figure 10. Our results indicate that Spotscape achieves fast training times, highlighting its practicality for high-throughput datasets (e.g., 100,000 spots) within a reasonable timeframe. Furthermore, to demonstrate the scalability of Spotscape for real-world,

large-scale datasets, we present the runtime for varying numbers of slices in Appendix G. These results show that Spotscape can handle extremely large datasets without an exponential increase in runtime as the number of slices grows.

## 6. Conclusion

In this work, we propose Spotscape, a novel framework for representation learning on SRT data, designed to address challenges in both single-slice and multi-slice tasks. Recognizing the limitations of relying solely on spatial locality due to the continuous nature of SRT data, Spotscape captures global spot similarities through the Similarity Telescope module, preserving a global similarity map invariant to augmentations. Additionally, we extend Spotscape to multi-slice tasks by employing a prototypical contrastive learning scheme and introducing a simple yet effective similarity scaling strategy to group spots with the same domain across different slices and mitigate batch effects. Extensive experiments demonstrate that Spotscape outperforms existing baselines, uncovering biologically meaningful insights and paving the way for more effective SRT analysis in diverse applications.

## Acknowledgements

This work was supported by Institute of Information & communications Technology Planning & Evaluation (IITP) grant funded by the Korea government(MSIT) (RS-2025-02304967, AI Star Fellowship(KAIST)), the National Research Foundation of Korea(NRF) grant funded by the Korea government(MSIT) (RS-2024-00406985), and the National Research Foundation of Korea(NRF) funded by Ministry of Science and ICT (RS-2022-NR068758).

## Impact Statement

This paper advances the field of Machine Learning for biosciences by improving spatial transcriptomics data analysis. Our proposed method may have potential applications in biomedical research, including disease understanding and precision medicine. The method may assist in identifying spatial gene expression patterns relevant to disease characterization or treatment development, but ethical considerations regarding data privacy and responsible use of machine learning in biomedical contexts should be considered in real-world applications. Beyond these general considerations, we think none require specific emphasis that require explicit discussion.

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

# A. Datasets

Table 5. Statistics for datasets used for experiments.

| Dataset | Species | Tissue | Technology | Resolution | Cells/Spots | Genes | # of Spatial Domains | Reference |
|---|---|---|---|---|---|---|---|---|
| DLPFC | Human | Brain (dorsolateral prefrontal cortex; DLPFC) | 10x Visium | 50 $\mu$m | 3460 ~ 4789 | 33538 | 5 ~ 7 | (Maynard et al., 2021) |
| MTG | Human | Brain (middle temporal gyrus; MTG) | 10x Visium | 50 $\mu$m | 3445 ~ 4832 | 36601 | 6 ~ 7 | (Chen et al., 2022b) |
| Mouse Embryo | Mouse | Whole embryo | Stereo-seq | 0.2 $\mu$m | 30756 ~ 55295 | 25485 ~ 27330 | 18 ~ 19 | (Chen et al., 2022a) |
| NSCLC | Human | Non-small cell lung cancer (NSCLC) | NanoString CosMX | Subcellular | 960 | 11756 | 4 | (Bhuva et al., 2024) |
| Breast Cancer | Human | Breast Cancer | 10x Visium | 50 $\mu$m | 4992 | 18085 | 11 | (Janesick et al., 2023) |
| Breast Cancer | Human | Breast Cancer | 10x Xenium | Subcellular | 167780 | 313 | 20 | (Janesick et al., 2023) |

In this section, we compare Spotscape with baseline methods on various datasets. The data statistics are in Table 5.

**Human Dorsolateral Prefrontal Cortex (DLPFC).** It comprises 12 tissue slices from 3 adult samples, with 4 consecutive slices per sample, derived from the dorsolateral prefrontal cortex. These slices were profiled using the 10x Visium platform. The original study manually annotated 6 neocortical layers (layers 1 to 6) as well as the white matter (see Figure 11).

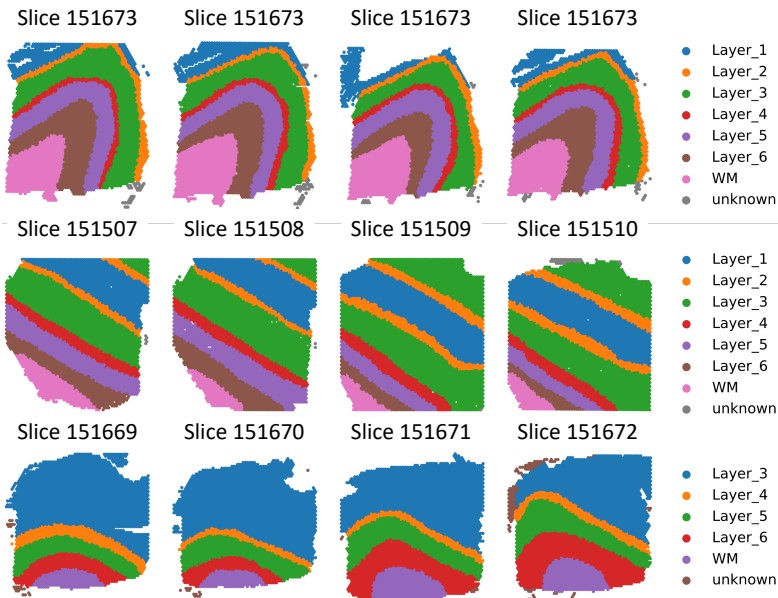

Figure 11. Spatial coordinates of DLPFC dataset.

**Middle Temporal Gyrus (MTG).** The MTG (middle temporal gyrus) dataset includes samples from both control and Alzheimer's disease (AD) groups. The MTG is a brain region particularly vulnerable to early AD pathology. In the original study, spatial transcriptomics profiles were characterized for both AD and control MTG samples by the 6 neocortical layers (layer 1 to 6) and white matter, utilizing the 10x Visium platform for detailed tissue profiling. The spot distribution is denoted in Figure 12.

**Mouse Embryo.** It is mouse whole embryo datasets by development stages. It was profiled by Stereo-seq technology, which allows spatial transcriptomics at the cellular level by integrating DNA nanoball-patterned arrays with in situ RNA capture. It offers a detailed spatiotemporal transcriptomic atlas (MOSTA) of mouse embryonic development (see Figure 14).

**Non-small Cell Lung Cancer (NSCLC).** The dataset comprises high-resolution, subcellular-level spatial transcriptomics data from human lung tissue, encompassing four distinct spatial domains (see Figure 13), including a tumor region. This data was generated using the NanoString CosMX platform.

**Human Breast Cancer**. It comprises spatial transcriptomics of human breast cancer tissues using 10x Visium for whole-transcriptome spatial data and 10x Xenium for high-resolution gene expression at the subcellular level. This combined approach offers detailed mapping of tumor microenvironments (see Figure 31), highlighting molecular differences and cell-type composition to better understand cancer heterogeneity and invasion.

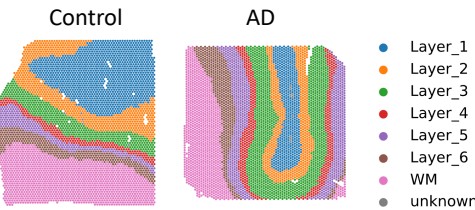

Figure 12. Spatial coordinates of MTG dataset.

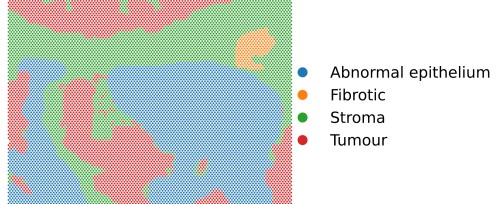

Figure 13. Spatial coordinates of NSCLC dataset.

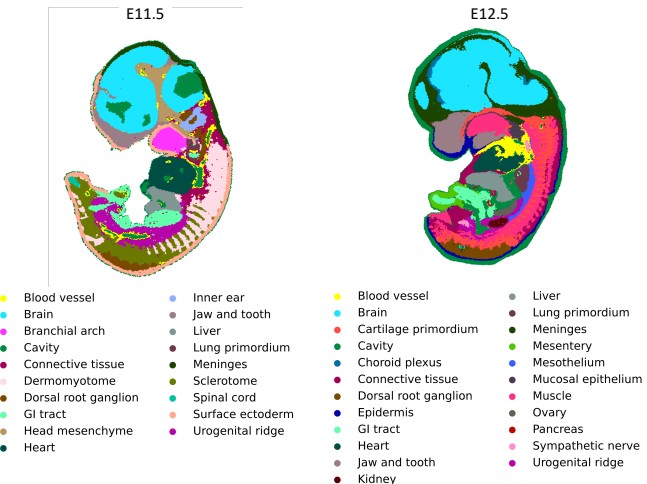

Figure 14. Spatial coordinates of Mouse Development dataset.

## B. Baseline Methods

In Table 6, we indicate which baseline methods are applicable to specific tasks, categorizing them based on whether their respective papers address those problems. Furthermore, we compare the performance of Spotscape with general self-supervised representation learning schemes. Graph Contrastive Learning (Chen et al., 2020; Zhu et al., 2020) is a instance-wise contrastive learning method that learns representations by pushing negative pairs apart and pulling positive pairs together. BGRL (Thakoor et al., 2021; Grill et al., 2020) is a consistency regularization method that learns representations by enforcing consistency between two differently augmented views. SwAV (Caron et al., 2020b) learns representations by minimizing the difference between two cluster assignments that are obtained through optimal transport. Barlow twins (Caron et al., 2020a) learns representations by minimizing redundancy between two augmented view. Although these methods demonstrate strong performance across various domains, our results in Figure 15 indicate that Spotscape is the most suitable model for SRT data, emphasizing its effectiveness in this context.

Table 6. Baseline methods and their application across various tasks.

| Method | Single-slice Tasks | | | Multi-slice Tasks | | | |
|---|---|---|---|---|---|---|---|
| | Spatial domain identification | Trajectory inference | Imputation | Homogeneous integration | Homogeneous alignment | Heterogeneous integration | Heterogeneous alignment |
| SEDR | ✔ | ✔ | ✔ | | | | |
| STAGATE | ✔ | ✔ | ✔ | | | | |
| SpaCAE | ✔ | ✔ | ✔ | | | | |
| SpaceFlow | ✔ | ✔ | | | | | |
| GraphST | ✔ | ✔ | ✔ | ✔ | | | |
| PASTE | | | | ✔ | ✔ | | |
| STAligner | | | | ✔ | ✔ | ✔ | ✔ |
| SLAT | | | | | ✔ | | ✔ |
| PASTE2 | | | | | ✔ | | ✔ |
| CAST | ✔ | ✔ | | ✔ | ✔ | ✔ | ✔ |
| Spotscape | ✔ | ✔ | ✔ | ✔ | ✔ | ✔ | ✔ |

Figure 15. Comparison with self-supervised learning.

## C. Pseudo Code

In this section, we provide pseudocode of Spotscape in Algorithm 1.

---

**Algorithm 1** Overall framework of Spotscape

---

**Require:** Spatial nearest neighbor graph $\mathcal{G} = (X, A)$, feature matrix $X$, adjacency matrix $A$, graph augmentation $\mathcal{T}$, GCN encoder $f_\theta$, MLP decoder $g_\theta$, number of slices $N_d$, number of spots $N_s$, number of latent dimensions $D$, loss balancing parameters $(\lambda_{\text{Recon}}, \lambda_{\text{SC}}, \lambda_{\text{PCL}}, \lambda_{\text{SS}})$, temperature $\tau$, learning rate $\eta$

**Ensure:** Node embeddings $Z$, reconstructed feature matrix $\hat{X}$

1: **for** epoch **in** epochs:
2:     $\tilde{\mathcal{G}} = \mathcal{T}(\mathcal{G})$, $\tilde{\mathcal{G}}' = \mathcal{T}'(\mathcal{G})$      /* two randomly augmented version of G */

3:     **Step 1: Graph Autoencoder**
4:     $\tilde{Z} = f_\theta(\mathcal{G})$, $\tilde{Z}' = f_\theta(\tilde{\mathcal{G}}')$      /* compute spot embedding using GCN encoder */
5:     $\hat{X} = g_\theta(\tilde{Z})$, $\hat{X}' = g_\theta(\tilde{Z}')$      /* reconstruct the feature matrix using MLP decoder */

6:     **Step 2: Similarity Telescope with Relation Consistency** (Section 4.2)
7:     $\mathcal{L}_{\text{Recon}} = \text{Reconstruction Loss}(X, \hat{X}, \hat{X}')$      (Eqn. 2)
8:     $\mathcal{L}_{\text{SC}}, H = \text{Similarity Telescope with Relation Consistency Loss}(\tilde{Z}, \tilde{Z}')$

9:     **Step 3: Prototypical Contrastive Learning** (Section 4.3)
10:     **if** $N_d \geq 2$ **and** epoch $\geq$ warm-up epoch **then**      /* for multi-slice only */
11:         $\mathcal{L}_{\text{PCL}} = \text{PCL Loss}(\tilde{Z}, \tilde{Z}')$
12:     **else**
13:         $\mathcal{L}_{\text{PCL}} = 0$
14:     **end if**

15:     **Step 4: Similarity Scaling Strategy** (Section 4.4)
16:     **if** $N_d \geq 2$ **then**      /* for multi-slice only */
17:         $\mathcal{L}_{\text{SS}} = \text{Similarity Scaling Loss}(H, \mathcal{G})$      (Eqn. 6, 7)
18:     **else**
19:         $\mathcal{L}_{\text{SS}} = 0$
20:     **end if**

21:     **Step 5: Compute Loss**
22:     $\mathcal{L} = \lambda_{\text{Recon}}\mathcal{L}_{\text{Recon}} + \lambda_{\text{SC}}\mathcal{L}_{\text{SC}} + \lambda_{\text{PCL}}\mathcal{L}_{\text{PCL}} + \lambda_{\text{SS}}\mathcal{L}_{\text{SS}}$

23:     **Step 6: Backpropagation and Parameter Update**
24:         Update parameters $\theta$ using Adam optimizer: $\theta_{\text{epoch}} \leftarrow \text{Adam}(\theta_{\text{epoch}-1}, \eta)$

25: **Return:** Node embeddings $Z$, reconstructed feature matrix $\hat{X}$

/* Utility Functions */
26: **Function** Similarity Telescope with Relation Consistency Loss$(\tilde{Z}, \tilde{Z}')$:
27:     $\tilde{Z}_{\text{norm}} = \text{L2-norm}(\tilde{Z})$, $\tilde{Z}'_{\text{norm}} = \text{L2-norm}(\tilde{Z}')$      /* L2-normalization */
28:     $H = \tilde{Z}_{\text{norm}}(\tilde{Z}'_{\text{norm}})^T$, $H' = \tilde{Z}'_{\text{norm}}(\tilde{Z}_{\text{norm}})^T$      /* compute cosine similarity */
29:     $\mathcal{L}_{\text{SC}} = \text{MSE}(H, H')$      (Eqn. 1)
30:     **Return:** $\mathcal{L}_{\text{SC}}, H$

31: **Function** PCL Loss$(\tilde{Z}, \tilde{Z}')$:
32:     # $P_{set}$: the collection of prototype sets from K-means clustering
33:     $P_{\text{set}} \leftarrow \text{Assign Prototype}(\tilde{Z}')$
34:     Calculate the prototypical contrastive loss $\mathcal{L}_{\text{PCL}}$ using $\tau$, $\tilde{Z}$, and $P_{\text{set}}$      (Eqn. 4, 5)
35:     **Return:** $\mathcal{L}_{\text{PCL}}$

36: **Function** Assign Prototype$(Z)$:
37:     $P_{\text{set}} \leftarrow [\,]$
38:     **for** $K$ **in** $[K^1, K^2, \ldots, K^T]$:
39:         Cluster each cell into $K$ clusters based on $Z$
40:         Compute a prototype matrix $P \in \mathbb{R}^{K \times D}$ by averaging of the spot embeddings per cluster
41:         Append $P$ to $P_{\text{set}}$
42:     **Return:** $P_{\text{set}}$

---

# D. Sensitivity Analysis

We conduct broad sensitivity analysis for all of the four balance parameters ($\lambda_{\text{Recon}}$, $\lambda_{\text{SC}}$, $\lambda_{\text{PCL}}$ $\lambda_{\text{SS}}$), temperature $\tau$, and the learning rate in Figure 16, 17, 18, and 19. Furthermore, the sensitivity to the number of clusters used in $K$-means clustering and the ground truth clusters is reported in Figure 20. These results indicate that the representation learned by Spotscape remains robust even when the number of clusters is not perfectly accurate with that ground truth clusters.

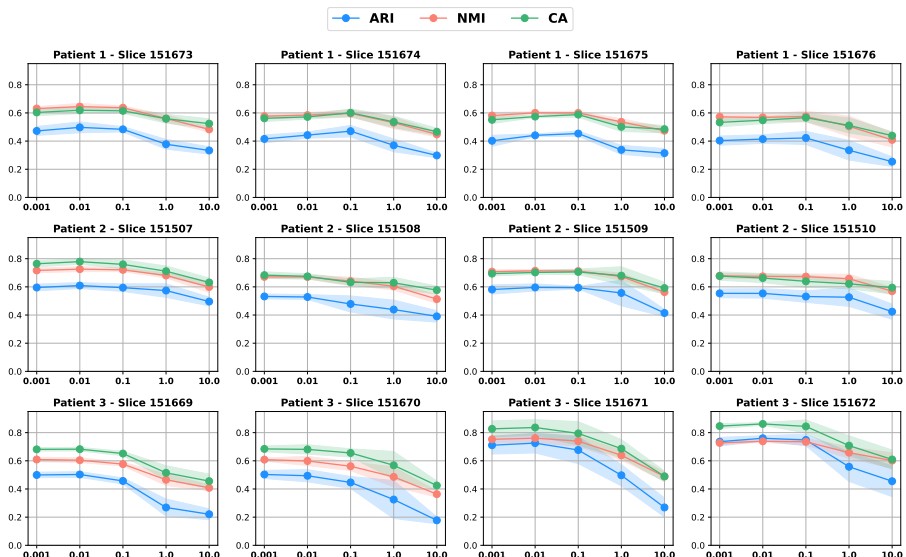

*Figure 16.* Sensitivity analysis for reconstruction loss balancing parameter ($\lambda_{\text{Recon}}$) of single DLPFC.

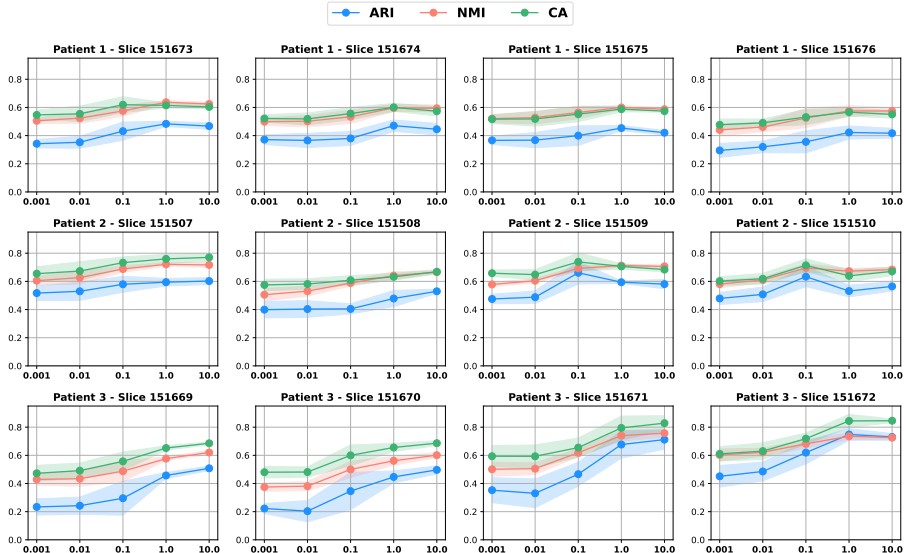

*Figure 17.* Sensitivity analysis for similarity telescope loss balancing parameter ($\lambda_{\text{SC}}$) of single DLPFC.

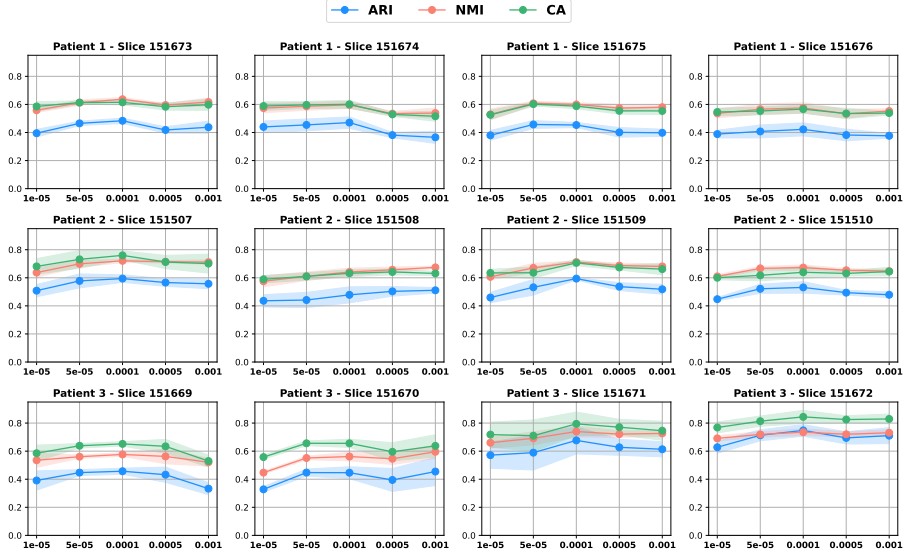

*Figure 18.* Sensitivity analysis for learning rate of single DLPFC.

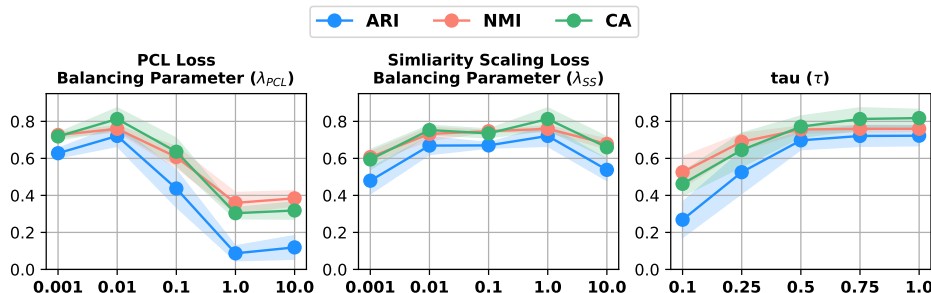

*Figure 19.* Sensitivity analysis for multi-slice parameters.

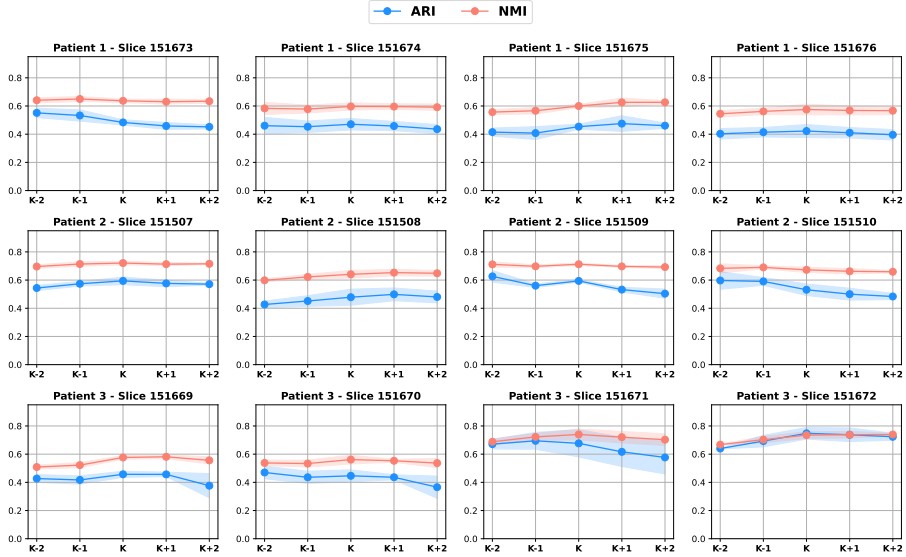

*Figure 20.* Sensitivity analysis for number of cluster (K) of single DLPFC.

# E. Hyperparameter Selection and Implementation Details

## E.1. Hyperparamter search for model performance comparison

To ensure a fair comparison, we conducted a hyperparameter search for both Spotscape and the baseline methods. The best-performing hyperparameters were selected by evaluating the NMI with the first seed. Specifically, for Spotscape, the hyperparameter search was conducted only for the learning rate, with the search space consisting of $\{0.00001, 0.00005, 0.0001, 0.0005, 0.001\}$. The remaining hyperparameters were fixed, and the ones used to report the experimental results are listed in Table 7.

*Table 7.* Hyperparameter settings of Spotscape.

| | Fixed | DLPFC Single | MTG Single | Mouse Embryo | NSCLC | DLPFC Multi Integration | MTG Multi Integration | Mouse Embryo Alignment | Visium - Xenium Alignment |
|---|---|---|---|---|---|---|---|---|---|
| $\lambda_{Recon}$ | ✔ | 0.1 | 0.1 | 0.1 | 0.1 | 0.1 | 0.1 | 0.1 | 0.1 |
| $\lambda_{SC}$ | ✔ | 1.0 | 1.0 | 1.0 | 1.0 | 1.0 | 1.0 | 1.0 | 1.0 |
| $\lambda_{PCL}$ | ✔ | N/A | N/A | N/A | N/A | 0.01 | 0.01 | 0.01 | 0.01 |
| $\lambda_{SS}$ | ✔ | N/A | N/A | N/A | N/A | 1.0 | 1.0 | 1.0 | 1.0 |
| GCN encoder dimensions | ✔ | $[N_g, 256, 64]$ | $[N_g, 256, 64]$ | $[N_g, 256, 64]$ | $[N_g, 256, 64]$ | $[N_g, 256, 64]$ | $[N_g, 256, 64]$ | $[N_g, 256, 64]$ | $[N_g, 256, 64]$ |
| $\tau$ | ✔ | N/A | N/A | N/A | N/A | 0.75 | 0.75 | 0.75 | 0.75 |
| Top-$k$ | ✔ | N/A | N/A | N/A | N/A | 5 | 5 | 5 | 5 |
| Training epochs | ✔ | 1000 | 1000 | 1000 | 1000 | 1000 | 1000 | 1000 | 1000 |
| Warm-up epochs | ✔ | 500 | 500 | 500 | 500 | 500 | 500 | 500 | 500 |
| Learning rate | ✘ | 0.0001 | 0.0001 | 0.00001 | 0.0005 | 0.0005 | 0.0005 | 0.001 | 0.00001 |
| Feature masking rate ($\mathcal{T}_f$, 1) | ✔ | 0.2 | 0.2 | 0.2 | 0.2 | 0.2 | 0.2 | 0.2 | 0.2 |
| Feature masking rate ($\mathcal{T}_f$, 2) | ✔ | 0.2 | 0.2 | 0.2 | 0.2 | 0.2 | 0.2 | 0.2 | 0.2 |
| Edge masking rate ($\mathcal{T}_e$, 1) | ✔ | 0.2 | 0.2 | 0.2 | 0.2 | 0.2 | 0.2 | 0.2 | 0.2 |
| Edge masking rate ($\mathcal{T}_e$, 2) | ✔ | 0.2 | 0.2 | 0.2 | 0.2 | 0.2 | 0.2 | 0.2 | 0.2 |

Additionally, we conducted a grid search primarily targeting the learning rate and loss balancing parameters for the baseline models. The learning rates for all baselines were explored within the search space $\{0.00001, 0.00005, 0.0001, 0.0005, 0.001, 0.005, 0.01, 0.05\}$. Similarly, the loss balancing parameters were tuned across the range $\{0.1, 1.0, 10.0\}$ including their default parameter. More precisely, for SEDR, it searched learning rate and balance parameters regarding reconstruction loss, VGAE loss, and self-supervised loss. For STAGATE, the search focused solely on the learning rate. In the case of SpaCAE, both the learning rate and the spatial expression augmentation parameter ($\alpha$) were tuned within $\{0.5, 1.0\}$. SpaceFlow was optimized by adjusting the learning rate and the spatial consistency loss balancing parameter. For GraphST, we explored the learning rate and the balancing parameters for feature reconstruction loss and self-supervised contrastive loss. Regarding STAligner, we searched for the optimal learning rates for both the pretrained model (i.e., STAGATE) and the fine-tuning process. Finally, for SLAT, we applied the default parameters since the experiments were conducted under identical settings and with the same dataset. This systematic parameter-tuning process facilitated the effective optimization of each baseline model's performance.

## E.2. Unsupervised Hyperparameter Search Strategy

*Table 8.* Optimized hyperparameter settings for Spotscape.

| Type | Dataset | Learning Rate |
|---|---|---|
| Single | DLPFC Patient 1 | 0.00005 |
| Single | DLPFC Patient 2 | 0.0001 |
| Single | DLPFC Patient 3 | 0.0001 |
| Single | MTG Control | 0.0005 |
| Single | MTG AD | 0.0001 |
| Single | Mouse Embryo | 0.00001 |
| Single | NSCLC | 0.00005 |
| Multi Integration | DLPFC | 0.001 |
| Multi Integration | MTG | 0.0005 |
| Multi Alignment | Mouse Embryo | 0.001 |
| Multi Alignment | Breast Cancer | 0.00001 |

To apply Spotscape to new data, an appropriate hyperparameter search strategy is essential. Fortunately, since Spotscape is largely robust to hyperparameters, we fix all parameters except the learning rate and search for the learning rate that maximizes the silhouette score, which can be achieved without any supervised information. Using this hyperparameter optimization strategy, we obtained the hyperparameters listed in Table 8 and reported the computed silhouette scores during the search process for DLPFC in Figure 21. It shows that the trend between the Silhouette Score and clustering performance (i.e., ARI) is similar, indicating that these strategies work well.

We then compared the performance of the hyperparameters optimized without supervision with that of the hyperparameters optimized with supervision, which were used solely for performance comparison with the baseline methods in Figure 22. In this comparison, the performances of both sets of hyperparameters are competitive, with the unsupervised optimization showing even better performance in some cases, thereby demonstrating the effectiveness of our search strategy and confirming the robustness of hyperparameter sensitivity.

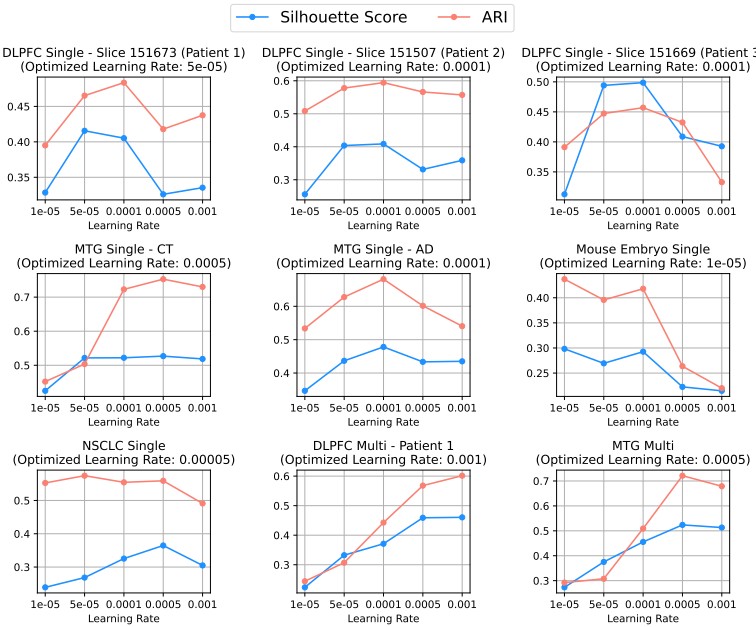

*Figure 21.* Unsupervised hyperparameter searching strategy using silhouette scores.

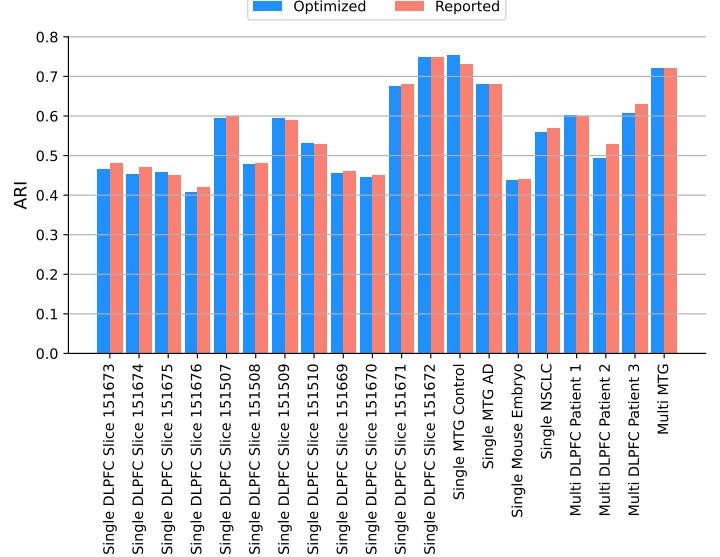

*Figure 22.* Performance comparison between optimized and reported hyperparameters.

### E.3. Implementation Details

**Model architecture and training.** The model employs a 2-layer GCN (Kipf & Welling, 2016) as the GNN-based encoder and a 2-layer MLP as the decoder, both utilizing batch normalization and ReLU activation functions. The encoder's hidden dimensions are set to $[N_g, 256, 64]$, while the decoder's dimensions are configured as $[64, 256, N_g]$. The clustering process in PCL is performed $T = 3$ times, with the $K$-means granularity set to $[K, 1.5K, 2K]$ to get a fine-grained representation.

Optimization is carried out using the Adam optimizer with a learning rate determined through hyperparameter searching (see Appendix E.1) and a weight decay of 0.0001.

**Preprocessing.** We follow the preprocessing methodology described in prior work (Dong & Zhang, 2022). Initially, 5000 highly variable genes are selected using Seurat v3 (Stuart et al., 2019). The data is then normalized to a CPM target of $10,000$ and log-transformed using the SCANPY package (Wolf et al., 2018). For datasets with multiple slices, we concatenate the slices to enable integration or alignment.

**Computational Resources.** All the experiments are conducted on Intel Xeon Gold 6326 CPU and NVIDIA GeForce A6000 (48GB).

**Software Configuration.** Spotscape is implemented in Python 3 (version 3.9.7) using PyTorch 2.1.1 (`https://pytorch.org/`) with Pytorch Geometric (`https://github.com/pyg-team/pytorch_geometric`) packages.

## F. Empirical Validation of Spotscape Architecture

**Loss design for single-slice tasks.** In this section, we evaluate the impact of PCL loss on single-slice tasks. While it can be applied in this setting, there is a trade-off between performance gains and computational cost, as shown in Figure 23. Although PCL loss helps cluster spots from the same spatial domain while separating others, the improvement is marginal compared to the significant increase in running time. Consequently, we chose to exclude PCL loss for single-slice tasks.

Additionally, similarity scaling loss, which is specifically designed to align similarities across different slices, is only applicable to multi-slice integration and does not apply to single-slice cases.

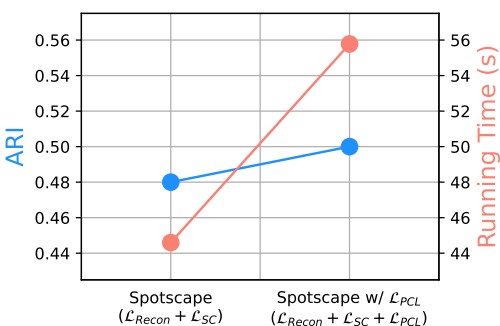

*Figure 23.* Trade-off between single-slice SDI performance and running time for $\mathcal{L}_{\text{PCL}}$

**GNN vs. MLP in Spatially Resolved Transcriptomics.** In this section, we aim to validate the architecture of Spotscape by investigating whether Graph Neural Networks (GNNs) are indeed more suitable than Multi-Layer Perceptrons (MLPs) in the domain of Spatially Resolved Transcriptomics. We further examine whether GNNs maintain their advantage over MLPs when modeling cell type-specific signals rather than spatial domain information.

To this end, we used the Postnatal Mouse Brain (PMB) from STOmicsDB (Xu et al., 2024b) (Database ID: STDS0000004), which is annotated by cell types. We conduct ablation studies comparing the performance of a GNN-based encoder with an MLP encoder on both the PMB dataset and the Dorsolateral Prefrontal Cortex (DLPFC) dataset, with clustering results reported in Tables 9 and 10.

Our results consistently show that the GNN-based encoder outperforms the MLP-based encoder for both spatial signals (i.e., DLPFC dataset) and cell type-specific signals (i.e., PMB dataset). However, the GNN-based encoder offers less pronounced benefits in the PMB dataset, which focuses on cell-type signals, compared to the DLPFC dataset, which emphasizes spatial domain clustering. This observation supports the intuition that spatial graphs encode richer spatial-specific information, while cell-type-specific signals are less spatially dependent. Nevertheless, the GNN still provides a measurable performance gain on the PMB dataset. This can be attributed to the fact that spatial regions inherently carry signals related to cell types, as similar cell types tend to cluster together within tissues. Such clustering reflects their functional and structural organization and their interactions within specific tissue regions. Supporting this, the homophily ratio for the Shared Nearest

Neighbor (SNN) graphs in both DLPFC and PMB datasets is high (0.92), indicating that spatial neighborhoods indeed contain substantial cell-type-related information.

*Table 9.* Spatial domain clustering performance for DLPFC dataset. **Bold** indicates the best performance, with the numbers in parentheses representing the standard deviation.

|  | ARI | NMI | CA |
|---|---|---|---|
| Spotscape (w/ MLP encoder) | 0.20 (0.01) | 0.30 (0.01) | 0.42 (0.02) |
| Spotscape | **0.48** (0.02) | **0.64** (0.01) | **0.61** (0.02) |

*Table 10.* Cell-type clustering performance for Postnatal Mouse Brain (PMB) dataset. **Bold** indicates the best performance, with the numbers in parentheses representing the standard deviation.

|  | ARI | NMI | CA |
|---|---|---|---|
| Spotscape (w/ MLP encoder) | 0.58 (0.03) | 0.65 (0.02) | 0.67 (0.03) |
| Spotscape | **0.61** (0.07) | **0.68** (0.03) | **0.74** (0.06) |

## G. Scalability of **Spotscape** for Large Dataset

In this section, we conduct additional analysis for time complexity using the mouse main olfactory bulb dataset from STOmicsDB (Xu et al., 2024b), which comprises 39 slices and a total of 1,792,797 spots. This result highlights the Spotscape's efficiency with extremely large-scale data, as shown in Figure 24.

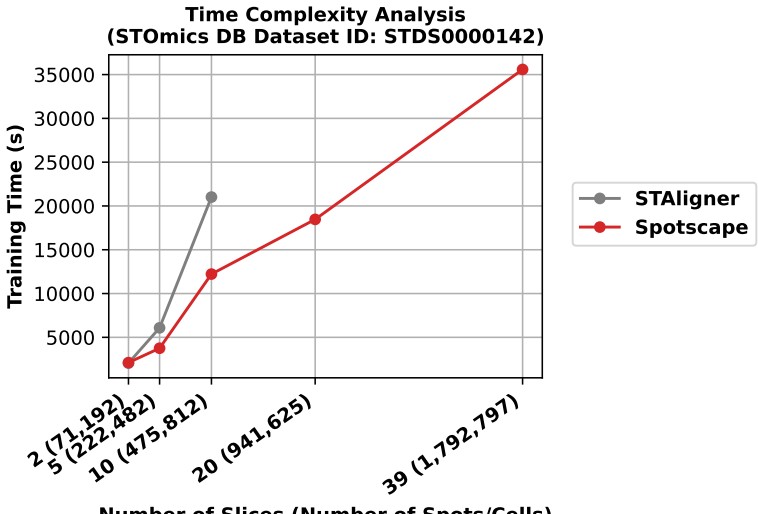

*Figure 24.* The running time of Spotscape and baseline methods over the various number of slices on the large dataset (ST Omics DB - Dataset ID: STDS0000142)

# H. Trajectory Analysis

For quantitative validation, we assign numerical values to layers as follows: WM = 0, layer 6 = 1, layer 5 = 2, layer 4 = 3, layer 3 = 4, layer 2 = 5, and layer 1 = 6. We then calculate pseudotimes following Spaceflow (Ren et al., 2022) using the representation from each model. Finally, we compute the spearman correlation with scaling between these assigned values and the calculated pseudotimes and report the results in Figure 25. We also present these results visually in Figure 26. In these results, Spotscape demonstrates effectiveness in the trajectory inference task, further validating its broad applicability.

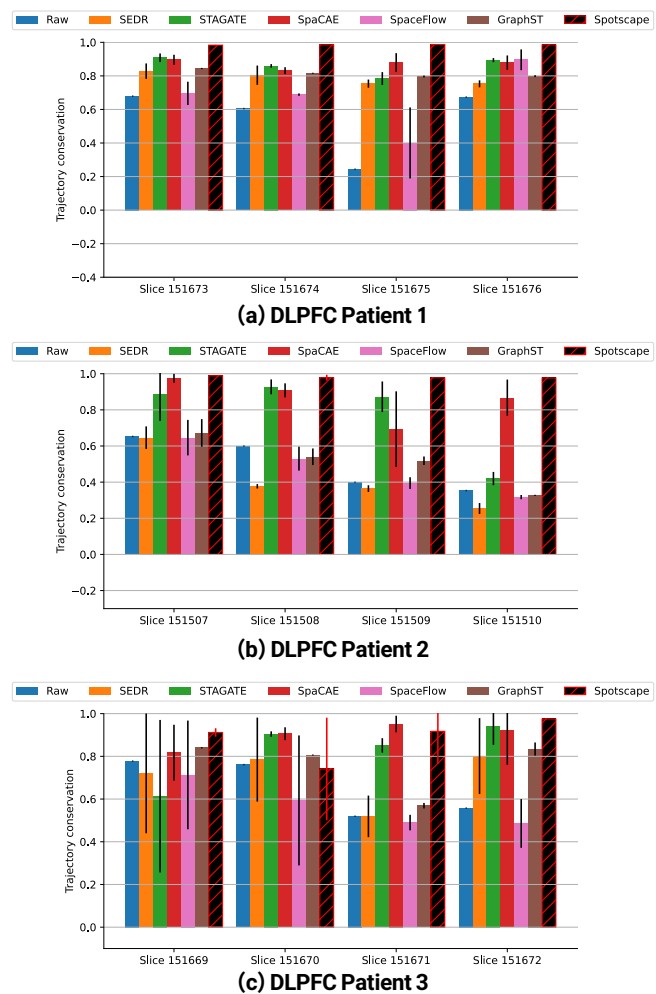

Figure 25. Trajectory conservation between pseudotimes and Layers in DLPFC data.

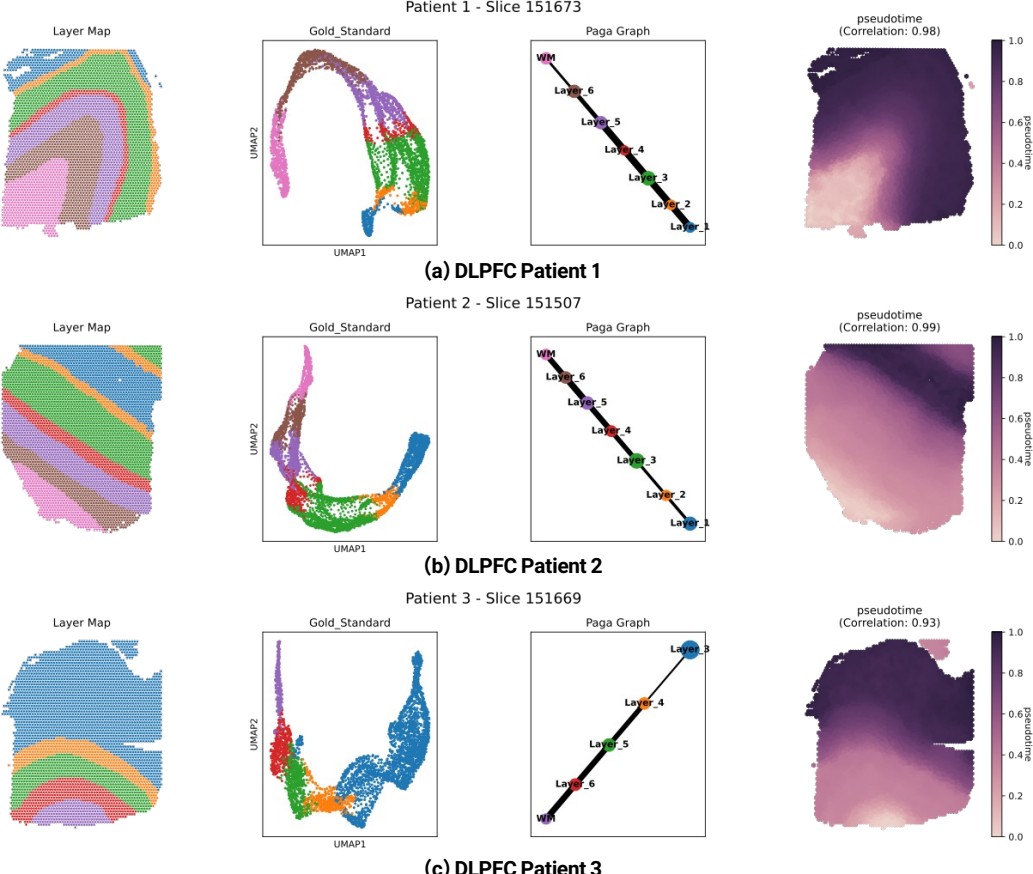

*Figure 26.* Trajectory inference results of Spotscape.

# I. Imputation

For quantitative evaluation with baseline methods, we masked certain non-zero values in the data and evaluated whether the model successfully recovers these values, following the settings from previous works (Lee et al., 2024). In Figure 27, Spotscape outperforms in terms of both RMSE and median L1-distance, demonstrating its superiority in imputation tasks. Moreover, we also report the broad results for marker gene detection results of denoised output from Spotscape in Figure 28. Specifically, RORB serves as a canonical marker for layer 4 neurons (Clark et al., 2020); ETV1 is associated with layer 5 neurons (Goralski et al., 2024); NTNG2 and NR4A2 are well-recognized markers for layer 6 neurons (Maynard et al., 2021; Darbandi et al., 2018); and OLIG2 is indicative of white matter regions (Wegener et al., 2015). The results show that after imputation using Spotscape, marker genes are more distinctly expressed, demonstrating the practical applicability of Spotscape.

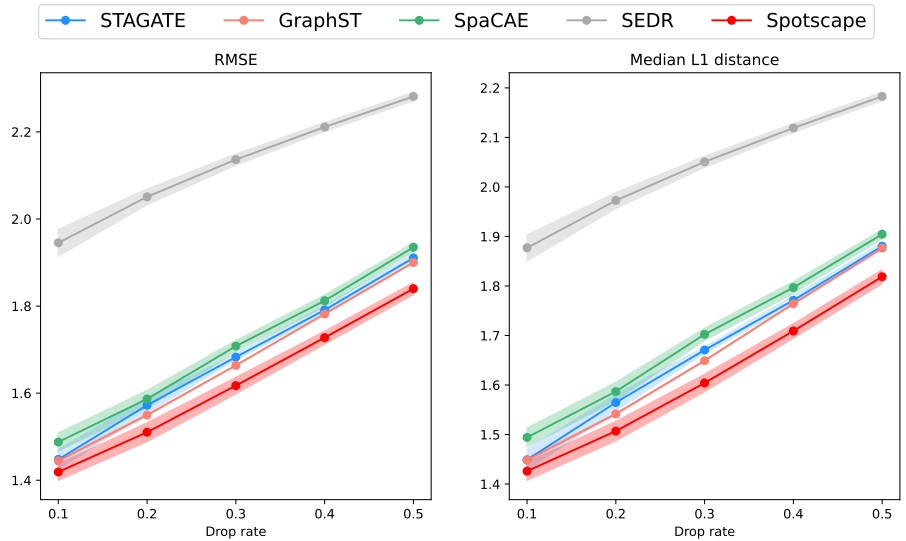

*Figure 27.* Imputation error comparison across various drop rates in the DLPFC.

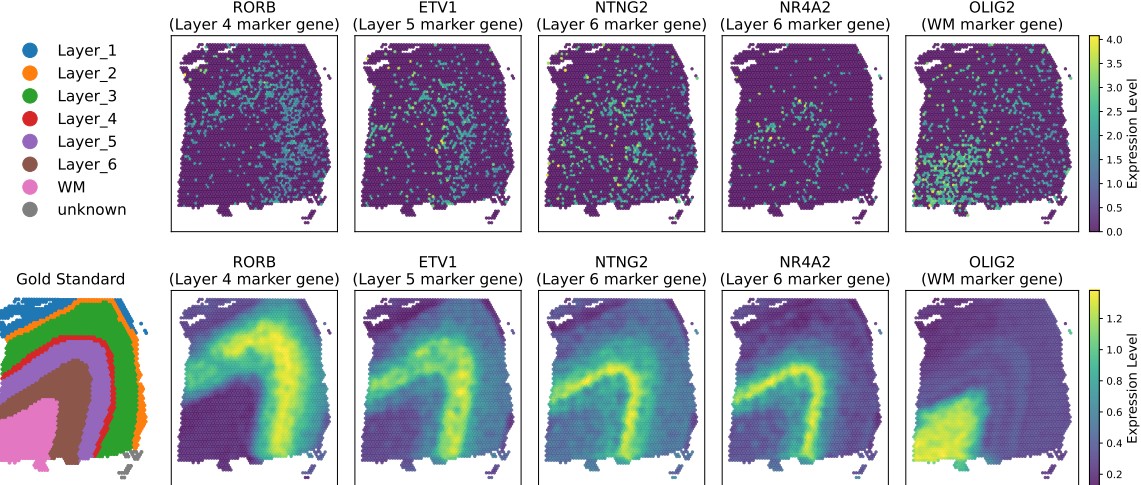

*Figure 28.* Spatial expression of raw and Spotscape imputed data for marker genes in DLPFC.

## J. Differentially Expressed Gene Analysis

To check whether our results yield biologically meaningful results, we investigate differentially expressed genes (DEGs) and their biological functions between the Control and Alzheimer's disease (AD) groups through Gene Ontology (GO) enrichment analysis for each cluster, representing a cortical layer in a brain. We use cells whose number of genes is between 500 and 7,500, the number of reads is between 1,000 and 30,000, and the ratio of the mitochondrial gene is below 35% for quality control. Genes with $\log_2(\text{fold}) > 0.25$ and an adjusted p-value of DESeq2 $< 0.05$ are considered differentially expressed genes (DEG).

Since Spotscape provides spatially organized and reliably distributed clusters as actual cortical layers in a brain, all clusters are assigned to the cortical layers. As the pathological influence of AD on different cortical layers is diverse, it is highly worthwhile to identify the differences between Control and AD in each region (Romito-DiGiacomo et al., 2007). We compare two clusters corresponding to layer 2 and layer 5, respectively. Layer 2 is regarded as a superficial layer, while layer 5 is deemed a deeper layer. As depicted in Figure 29 and Figure 30, terms relevant to AD such as regulation of apoptosis, microglial cell activation, and synapse pruning are enriched in both layer 2 and layer 5, identifying the shared alteration of AD and ensuring the reliability of the result (Goel et al., 2022; Gao et al., 2023; Brucato & Benjamin, 2020). Comparison of the top terms enriched uniquely in each layer provides interesting observations, that is, layer 2 shows hallmark processes of early-stage AD pathology, such as oxidative stress responses, while layer 5 reflects later-stage events involving inclusion body assembly and advanced apoptotic pathways. Critically, these findings align with the established understanding that layer 2 is among the earliest sites impacted in AD, whereas deeper layers (including layer 5) exhibit more pronounced synaptic and proteostatic perturbations in later stages (Romito-DiGiacomo et al., 2007). By delineating clusters that map onto these distinct laminar features, Spotscape demonstrates utility in uncovering meaningful biological variation from spatial transcriptomics data and in corroborating the different influences of AD across cortical layers.

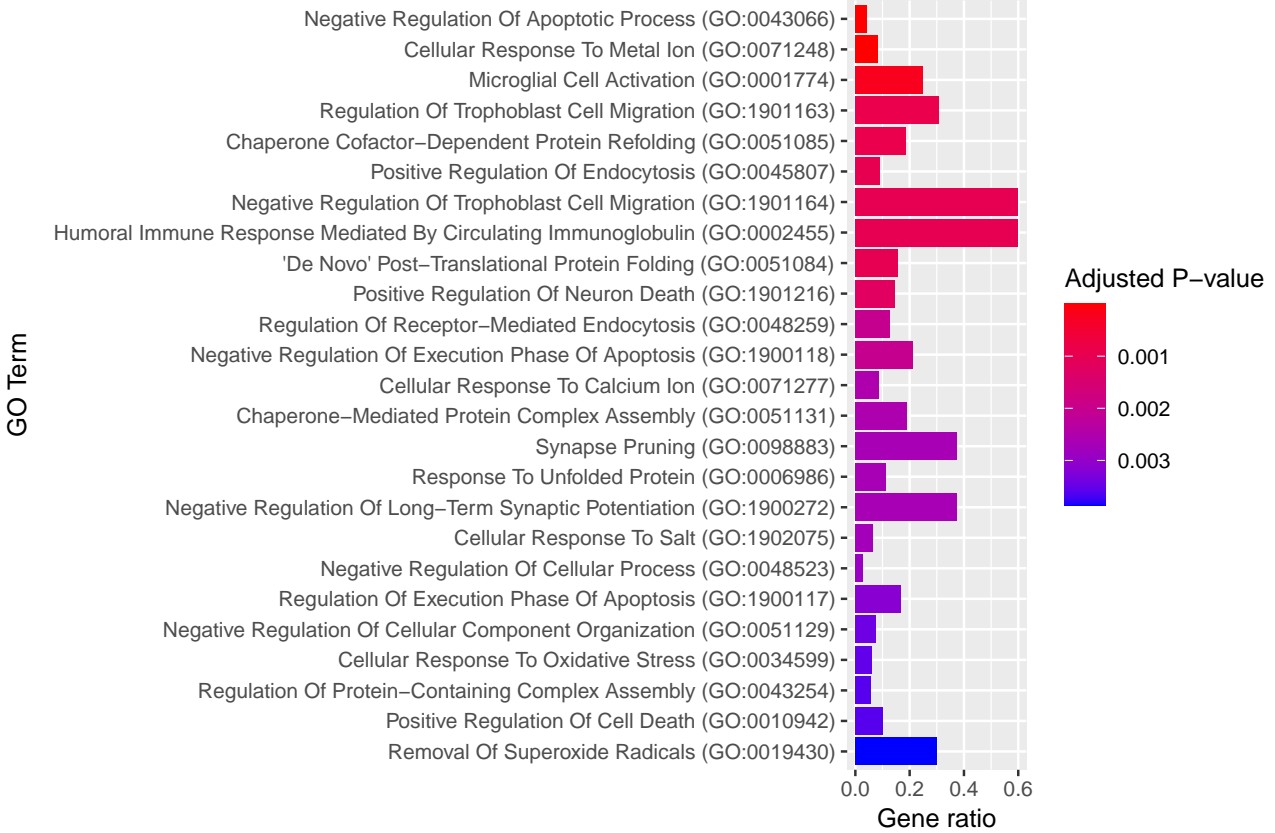

*Figure 29.* Top 25 biological process that DEGs between AD and Control enriched in a cluster assigned to layer 2.

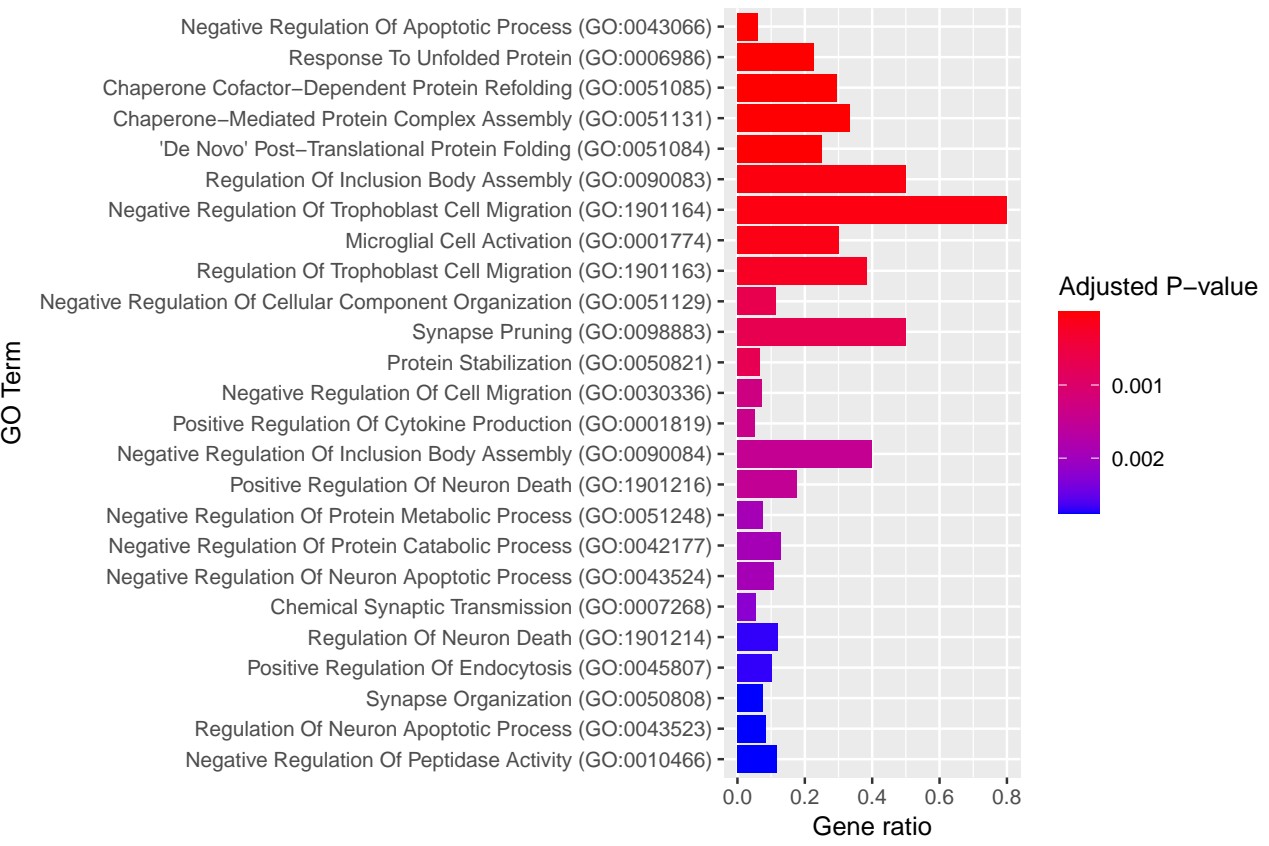

*Figure 30.* Top 25 biological process that DEGs between AD and Control enriched in a cluster assigned to layer 5.

## K. Cross-technology Alignment

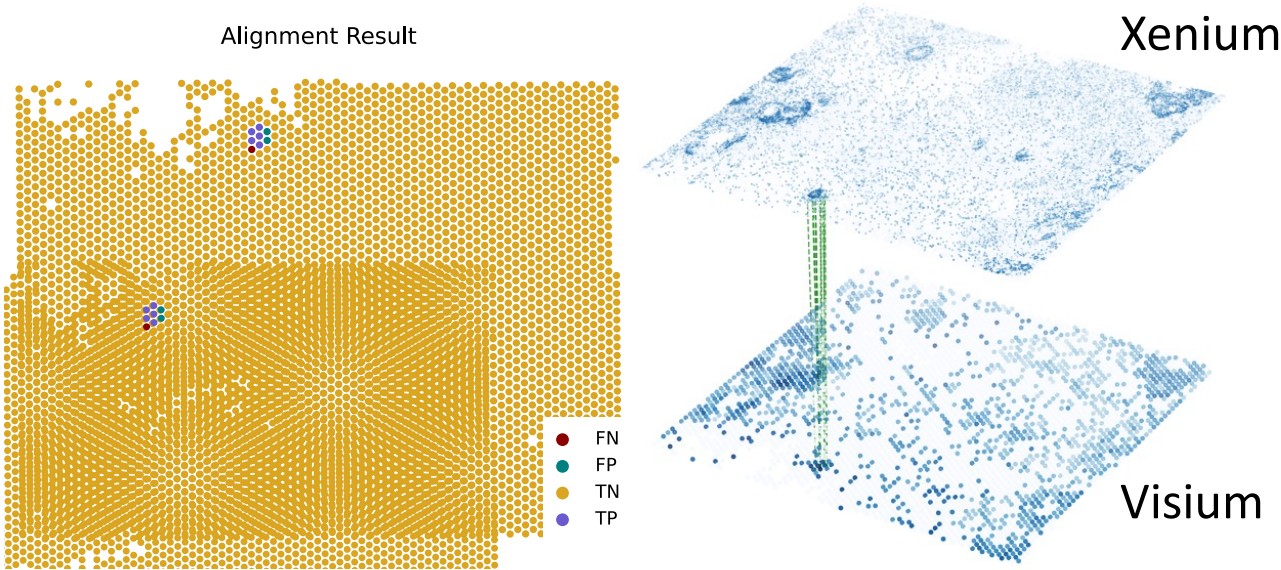

Figure 31. Alignment results of triple positive cells.

We conduct cross-technology alignment between data obtained from Xenium and Visium. Since Xenium offers higher resolution than Visium, while Visium provides a more comprehensive transcriptome view, aligning Xenium with Visium creates a complementary approach that combines the strengths of both: high resolution and broader coverage. To this end, we align triple-positive cells in Xenium—those positively enriched for the ERBB2, PGR, and ESR1 marker genes associated with breast tumors—with corresponding Visium spots. Figure 31 shows that Spotscape successfully outputs seven aligned points and identifies five triple-positive cells in the Visium data. This demonstrates the superiority of Spotscape, as it can successfully align extremely rare cell types (e.g., cancer cells).

## L. Interpretation of $\mathcal{L}_{\text{SC}}$

The similarity consistency loss in Equation (1) serves two key purposes: **1)** explicitly guiding the embedding space to capture quantitative similarity relationships, and **2)** encouraging the model to learn a global relational structure that spans all nodes.

In downstream tasks, we rely on cosine similarity between normalized embeddings. In other words, we treat the normalized embedding space as an Euclidean space, where distance serves as a proxy for semantic closeness. However, embeddings from GAE trained solely with the reconstruction loss are optimized for compression. As a result, the cosine similarity between embeddings lacks direct interpretability or consistency across different pairs. In an Euclidean space, the consistency loss equals zero ideally. Thus, the consistency loss serves as a regularizer that aligns the embedding space more closely with the desired geometry, making similarity values more meaningful and consistent.

Moreover, the reconstruction loss satisfies $\frac{\partial^2 \mathcal{L}_{\text{Recon}}}{\partial \tilde{Z}_i \partial \tilde{Z}'_j} = 0$ for $i \neq j$, which implies that updates to each embedding do not depend on the others. This is problematic, since downstream tasks involve comparing representations of even distant nodes. The consistency loss helps mitigate this limitation by considering similarity relations between all node pairs. The second derivative of the consistency loss, $\frac{\partial^2 \mathcal{L}_{\text{SC}}}{\partial \tilde{Z}_i \partial \tilde{Z}'_j}$, can have non-zero values, indicating that the update to each embedding depends on others. A detailed proof is provided below.

Let $P = \tilde{Z}_{\text{norm}}(\tilde{Z}'_{\text{norm}})^T, \quad Q = \tilde{Z}'_{\text{norm}}(\tilde{Z}_{\text{norm}})^T$.

Since $P = Q^T$,

$$\frac{\partial \mathcal{L}_{\text{SC}}}{\partial \tilde{z}_k} = \frac{2}{N_s^2} \sum_{i,j} (P_{ij} - Q_{ij})\left(\frac{\partial P_{ij}}{\partial \tilde{z}_k} - \frac{\partial Q_{ij}}{\partial \tilde{z}_k}\right) = \frac{4}{N_s^2} \sum_{i,j} (P_{ij} - Q_{ij})\frac{\partial P_{ij}}{\partial \tilde{z}_k}. \tag{9}$$

In addition,

$$\frac{\partial P_{kj}}{\partial \tilde{z}_k} = \frac{1}{||z_k||}(I - \frac{1}{||\tilde{z}_k||^2}\tilde{z}_k\tilde{z}_k^T)\frac{\tilde{z}_j{}'}{||\tilde{z}_j{}'||}, \tag{10}$$

$$\frac{\partial P_{ij}}{\partial \tilde{z}_k} = 0, \qquad\qquad\qquad \text{if } i \neq k. \tag{11}$$

Hence,

$$\frac{\partial \mathcal{L}_{\text{SC}}}{\partial \tilde{z}_k} \propto \sum_j (P_{kj} - Q_{kj})\frac{\partial P_{kj}}{\partial \tilde{z}_k} = \sum_j (P_{kj} - Q_{kj})\frac{1}{||\tilde{z}_k||}(I - \frac{1}{||\tilde{z}_k||^2}\tilde{z}_k\tilde{z}_k^T)\frac{\tilde{z}_j{}'}{||\tilde{z}_j{}'||}. \tag{12}$$

Therefore, Eq (13) can take nonzero values.

$$\frac{\partial^2 \mathcal{L}_{\text{SC}}}{\partial \tilde{z}_j'\partial \tilde{z}_k} \propto (P_{kj} - Q_{kj})\frac{1}{||\tilde{z}_k|| \cdot ||\tilde{z}_j'||}(I - \frac{1}{||\tilde{z}_k||^2}\tilde{z}_k\tilde{z}_k^T) \tag{13}$$

In summary, the consistency loss not only enhances the quantitative interpretability of similarity in the learned space, but also provides a mechanism for global information flow, thus improving the utility of the representations for downstream tasks.

## M. Future Works

In this work, we discover that reflecting the global relationships between spots provides significant information on SRT data; however, we currently leverage this relationship only implicitly through the loss function. We recognize that the model could benefit from incorporating more complex interactions by constructing edges between spots, thereby implementing graph structure learning. Future work could explore this avenue to enhance the representation of spatial relationships, allowing the model to leverage valuable information from the global context more effectively.

Furthermore, SRT data frequently includes histology images that offer critical contextual information about tissue architecture and cellular organization. However, in this study, we concentrate on a more general case that limits our analysis to spatial coordinates and gene expression profiles, potentially overlooking the rich insights that histological features could provide. We anticipate that integrating this information with Spotscape could represent a promising direction for future research.

