# OpenReview forum: "Global Context-aware Representation Learning for Spatially Resolved Transcriptomics"
_ICML.cc/2025/Conference — ICML 2025 poster_

### Official Review · Reviewer_aapZ · 2025-02-16

**Overall Recommendation:** 3

**Summary:**

The paper introduces Spotscape, a novel framework for representation learning in Spatially Resolved Transcriptomics (SRT) data. The key contribution of the paper is the Similarity Telescope module, which captures global relationships between spots, addressing the limitations of existing graph-based methods that rely heavily on local spatial information. Additionally, the paper extends Spotscape to multi-slice tasks by introducing a prototypical contrastive learning (PCL) scheme and a similarity scaling strategy to mitigate batch effects during multi-slice integration. The authors conduct extensive experiments on multiple datasets, demonstrating the superiority in various downstream tasks.

**Claims And Evidence:**

The claims made in the paper are generally well-supported by clear and convincing evidence. The authors provide extensive experimental results across multiple datasets, showing that Spotscape outperforms existing baselines in tasks such as spatial domain identification, trajectory inference, and multi-slice integration. The ablation studies further validate the importance of each proposed module (e.g., Similarity Telescope, PCL, and similarity scaling). However, the paper could benefit from a more detailed discussion on the theoretical justification for the global similarity learning scheme, particularly how it addresses the limitations of local spatial information in SRT data.

**Essential References Not Discussed:**

The paper adequately covers the relevant literature, but it could benefit from a discussion of recent advances in SRT clustering and multi-view representation learning, which are closely related to the proposed method.

**Experimental Designs Or Analyses:**

The experimental design is sound and well-executed. The authors evaluate Spotscape on multiple datasets, covering both single-slice and multi-slice scenarios. The results are consistent across different datasets, demonstrating the generalizability of the proposed method. The ablation studies and sensitivity analysis provide valuable insights into the contribution of each module. However, the paper could benefit from a more detailed discussion of the limitations of the proposed method, particularly in scenarios where the global similarity assumption may not hold.

**Methods And Evaluation Criteria:**

The proposed methods, including the Similarity Telescope module, prototypical contrastive learning (PCL) scheme, and similarity scaling strategy, are designed to address key challenges in Spatially Resolved Transcriptomics (SRT) data analysis. While the authors claim that these modules are novel and effective, I find that the theoretical justification and logical coherence of these methods are not sufficiently robust. For the Similarity Telescope Module, the authors argue that capturing global relationships between spots is crucial, especially for spots near spatial domain boundaries. However, the paper lacks a clear theoretical foundation for why global similarity learning is superior to local spatial information in all cases.

**Other Comments Or Suggestions:**

The paper is well-written presents the contributions and results. However, the authors should consider adding a discussion on the limitations of their method, particularly in scenarios where the global similarity assumption may not hold.

**Other Strengths And Weaknesses:**

Strengths:
1)The extension to multi-slice tasks using prototypical contrastive learning is well-designed and addresses a key challenge in SRT analysis.
2)The extensive experimental results and ablation studies provide strong evidence for the effectiveness of the proposed method.
3) The code is made publicly available, which enhances the reproducibility of the results.
Weaknesses:
1) The paper lacks a formal theoretical analysis of the global similarity learning mechanism.
2) The limitations of the proposed method, particularly in scenarios where the global similarity assumption may not hold, are not thoroughly discussed. For example, considering the relationship between spots from a global perspective may introduce noise, making it difficult to learn spot information with the same semantic relationship.

**Questions For Authors:**

1) Theoretical Justification:Could the authors provide a more formal theoretical justification for the global similarity learning mechanism? How does it address the limitations of local spatial information in SRT data?
2) What are the limitations of the proposed method, particularly in scenarios where the global similarity assumption may not hold? How might these limitations be addressed in future work? For example, considering the relationship between spots from a global perspective may introduce noise, making it difficult to learn spot information with the same semantic relationship.
3) In the comparison methods, please add the latest SRT clustering methods, such as [MAFN, TKDE'24] and [stGCL, ACM MM'24], to further validate the effectiveness of the proposed method.
4) The results indicate that Spotscape achieves fast training times, highlighting its practicality for high-throughput datasets (e.g., 100,000 spots) within a reasonable timeframe. However, the introduction of the prototypical contrastive learning (PCL) scheme might slow down the training process. Could the authors provide an analysis of the time complexity of each module, particularly focusing on the impact of PCL on the overall training time? This would help clarify the trade-offs between performance gains and computational cost.

**Relation To Broader Scientific Literature:**

The authors acknowledge the limitations of existing graph-based methods and propose a novel approach to address these limitations. The introduction of global similarity learning and prototypical contrastive learning aligns with recent trends in self-supervised learning and graph representation learning. The paper builds on prior work such as STAGATE and SpaceFlow, but introduces significant improvements by incorporating global context and multi-slice integration.

**Theoretical Claims:**

The paper does not present formal theoretical proofs. The authors provide intuitive explanations for the design choices, such as the global similarity learning scheme and the prototypical contrastive learning module. While the lack of theoretical guarantees is not a major drawback, a more formal analysis of the global similarity learning mechanism could strengthen the paper.

---

> ### Author Rebuttal · Authors · 2025-04-01
>
> Thank you for taking the time to provide constructive feedback on our paper. To address your concerns, we have added tables and figures in this [external link](https://anonymous.4open.science/r/Spotscape-31B6/Rebuttal.pdf)
>
> **Q1) Theoretical Justification**
>
> The similarity consistency loss in _Equation (1) of our manuscript_ serves two key purposes: **1) explicitly guiding the embedding space to capture quantitative similarity relationships**, and **2) encouraging the model to learn a global relational structure that spans all nodes**.
>
> In downstream tasks, we rely on cosine similarity between normalized embeddings. In other words, we treat the normalized embedding space as an Euclidean space, where distance serves as a proxy for semantic closeness. However, embeddings from GAE trained solely with the reconstruction loss are optimized for compression. As a result, the cosine similarity between embeddings lacks direct interpretability or consistency across different pairs. In an Euclidean space, the consistency loss equals zero ideally. Thus, the consistency loss serves as a regularizer that aligns the embedding space more closely with the desired geometry, making similarity values more meaningful and consistent.
>
> Moreover, the limited receptive field of GNNs is problematic because downstream tasks involve comparing representations of distant nodes. The consistency loss helps mitigate this limitation by considering similarity relations between all node pairs. The reconstruction loss satisfies $\frac{\partial^2 \mathcal{L}_{\text{recon}}}{\partial \tilde{Z}_i \partial \tilde{Z'}_j} = 0$ for $i \neq j$, which implies that updates to each embedding do not depend on the others.
>
> In contrast, the second derivative of the consistency loss, $\frac{\partial^2 \mathcal{L}_{\text{SC}}}{\partial \tilde{Z}_i \partial \tilde{Z'}_j}$, can have non-zero values, indicating that the update to each embedding depends on others as shown in our proof **R1.1 in the external link**.
>
> In summary, the consistency loss not only enhances the quantitative interpretability of similarity in the learned space, but also provides a mechanism for global information flow, thus improving the utility of the representations for downstream tasks.
>
> **Q2) Scenarios where the global similarity assumption may not hold**
>
> We understand your concern that Spotscape could introduce noise, particularly when spots far from the anchor node but still within the same spatial domain are considered.
> However, we want to clarify that our method does not rely on any form of aggregation from global nodes. Instead, Spotscape is designed to learn the similarities between all spots, irrespective of their global or local relationships.
>
> Our approach focuses on learning these similarities directly, without assuming that distant spots within the same spatial domain must always share meaningful information. Rather than aggregating information from any nodes, the model learns the relationships between spots through augmentation-based consistency, which expects that only meaningful relationships are captured. This avoids the risk of noise from distant spots and emphasizes learning the inherent semantic relationships between spots based on the data itself.
>
> In summary, Spotscape prioritizes the direct learning of spot similarities and consistency across augmentations, without aggregating information from distant or global nodes, which could potentially introduce noise.
>
> **Q3) More baselines**
>
> We have updated our experiments to include these two latest SRT clustering methods, MAFN (TKDE'24) and stGCL (ACM MM'24), across all relevant tasks, including clustering and integration. The results are now reported in **Tables 1, 2, and 3 in the external link**.
>
> Compared to these baselines, Spotscape still demonstrates superior performance, highlighting the effectiveness of our proposed method.
>
> **Q4) Trade-offs between performance gains and computational cost of the PCL scheme**
>
> To address your concern regarding the trade-offs between performance gains and computational costs associated with prototypical contrastive learning (PCL), we generated a synthetic dataset using scCube [1], with mouse embryo data as the reference, and reported the results in **Figure 6 (external link)**. This analysis demonstrates that while PCL does require more training time, it leads to better performance.
> We argue that in practical scenarios, the decision to use PCL depends on the user's preference for balancing training time and performance. However, we emphasize that PCL does not lead to impractical training times, as it is not excessively slow, and can still be viable for real-world applications.
>
> ---
> [1] "Simulating multiple variability in spatially resolved transcriptomics with scCube," Nature Communications

---

### Official Review · Reviewer_eTFY · 2025-03-02

**Overall Recommendation:** 2

**Summary:**

This paper proposed a new computational method, known as Spotscape, to integrate different spatial transcriptomics data. This model is improved by graph neural networks.

## update after rebuttal

I keep my score.

**Claims And Evidence:**

Yes.

**Essential References Not Discussed:**

NA.

**Experimental Designs Or Analyses:**

Yes, I have checked the analyses and designs. I have some questions about this, which is discussed in my later question section.

**Methods And Evaluation Criteria:**

Yes.

**Other Comments Or Suggestions:**

Please see my questions.

**Other Strengths And Weaknesses:**

Please see my comments.

**Questions For Authors:**

This paper proposed a method for spatial transcriptomics integration, which seems to have good performance, but I have some questions about benchmarking analysis and model design. If the authors can address my concerns, I may consider raising my score.

1. The presentation of Figure 1 is not clear to me. It seems that GAE variations and the proposed method identify similar regions. Why proposed method cannot exceed GAE variation in both metrics? What is the meaning of the colors presented in the right panels? I think it does not align well with the labels presented in the left panel. The authors should consider improving the presentation.

2. It has been shown that the variation of spatial transcriptomic data can be decomposed into cell type-specific signals and spatial signals. Therefore, is it still meaningful to use a GNN to encode cell-type specific signals? The answers should include oblivion studies based on normal MLP.

3. What about the scalability of the purposed models can it handle large-scale datasets? For example, the spatial atlas data suggested by spatial AD/HC database: https://ngdc.cncb.ac.cn/databasecommons/database/id/9046 or STImage 1K4M (https://github.com/JiawenChenn/STimage-1K4M)?

4. The authors do not present the turning results of baseline methods. Do they keep the default hyper-parameters for the baselines? If so, how to ensure the comparison is fair as the authors are tuning their proposed method across different datasets? Also, the authors may need to consider PASTE2 (https://github.com/raphael-group/paste2) or CAST (https://www.nature.com/articles/s41592-024-02410-7) as a baseline.

5. The UMAP presented in Figure 6 looks strange. Do they observe trajectory patterns for this figure? If so, it will be helpful if they also compute scores based on the raw expression profiles as I think the raw data capture the most information on trajectory relationship and it looks like we do not need to run other methods in analyzing this dataset.

**Relation To Broader Scientific Literature:**

The contribution is minor, but the scientists working on spatial transcriptomics analysis will be interested in reading it.

**Theoretical Claims:**

Yes, I have checked the correctness of proofs and claims.

---

> ### Author Rebuttal · Authors · 2025-04-01
>
> Thank you for your constructive feedback. To address your concerns, we upload additional results in the [external link](https://anonymous.4open.science/r/Spotscape-31B6/Rebuttal.pdf).
>
> **Q1-1) Presentation of Figure 1**
>
> _Figure 1_ highlights the limitations of previous methods that address the issues present in the Vanilla GAE (b-1), particularly the problem where boundary spots receive noisy information from spots in different spatial domains.
>
> While we describe the observation from _Figure 1 in the second paragraph of our manuscript_, we understand that the presentation of _Figure 1_ may have caused some confusion, as we did not clearly indicate which part of _Figure 1_ readers should focus on. The most important part of _Figure 1_ is the number of red dots, which represent **Wrongly Clustered in Boundary** and **boundary CA**. We will revise the manuscript to make this point more prominent.
>
> Additionally, the spot colors in _Figure 1_ (a) and (b) are independent. Please refer to the legend at the top of the _Figure 1_ (a) of our manuscript.
>
> **Q1-2) Spotscape vs. GAE variation**
>
> Spotscape did not exceed GAE with **oracle edges** (b-3), which is connected with the same ground truth only, in terms of Total CA, which is not a real scenario. However, Spotscape does outperform the oracle setting in terms of boundary CA, demonstrating the effectiveness of our proposed method in addressing the boundary-related issues.
>
> **Q2) GNN for cell-type specific signals**
>
> Your concern arises from the fact that all of the datasets we used are annotated by spatial domains rather than cell types, and that our evaluations only focus on spatial domains. You may also have assumed that as our graph is primarily constructed based on spatial coordinates, this would be only effective for capturing spatial signals, while not as useful for encoding cell-type-specific signals.
>
> To address this, we introduce a new dataset from the **Postnatal Mouse Brain (PMB)** (STOmicsDB ID: STDS0000004), **annotated by cell types**, and perform ablation studies with an MLP encoder. The clustering performance for PMB and DLPFC both is reported in **Table 5 and 6 in the external link**.
>
> We observe that the GNN-based encoder is less beneficial in the PMB dataset (cell-type) than in the DLPFC dataset (spatial domain) clustering. This indeed aligns with your intuition that the spatial graph carries more information regarding spatial-specific signals rather than cell-type-specific signals. However, GNN still provides a performance gain in the PMB data since spatial regions also implicitly carry signals related to cell types because the same cell types tend to cluster together within tissues, reflecting their functional and structural organization, as well as their interactions within specific tissue regions [1]. In fact, the homophily ratio of both the SNN graph in the DLPFC and the PMB data is 0.92, demonstrating that spatial regions also carry cell-type-related signals.
>
> **Q3) Scalability**
>
> Please refer to _Figure 10 in our manuscript_ about reasonable runtime up to 100,000 spots. To further address your concern, we conducted additional experiments using the Mouse Main Olfactory Bulb dataset (STOmicsDB ID: STDS0000142 - 1,792,797 spots in 39 slices).
> In **Figure 5 in the external link**, we present the runtime for different numbers of slices, demonstrating that Spotscape can handle large datasets without exponential runtime growth as the number of slices grows.
>
> **Q4-1) Hyper-parameters setting**
>
> Please refer to our rebuttal regarding **Q4** to reviewer **uFL9** due to the character limits.
>
> **Q4-2) More baselines**
>
> We have updated the baselines **in the external link**.
> * PASTE2: Slice-to slice alignment using optimal transport in **Table 4**
> * CAST: Integration and alignment using CCA-SSG in **Table 3** (Heterogeneous integration), **Figure 1** (UMAP for batch effect), **Figure 3** (Trajectory inference), and **Table 4** (Alignment)
>
> **Q5-1) Figure 6 UMAP & Trajectory of raw expression**
>
> We did not intend to show trajectory inference results in _Figure 6_. Our goal was to highlight the performance of the heterogeneous integration reducing batch effect.
>
> To address your concern, we show that the scores based on the raw expression do not provide reliable information on trajectory. To evaluate this, we calculate the pseudo-Spatiotemporal Map and report its correlation with the ground-truth layer, adding the results in **Figure 2 in the external link**. These results indicate that raw expression alone performs the worst, demonstrating the necessity of additional representation learning methods.
>
> Finally, we also conducted trajectory inference on this dataset and reported the results **in Figure 3 and 4 in the external link**. These results show that while Spotscape successfully captures the trajectory, the raw expression-based approach fails to do so.
>
> ---
> [1] "An introduction to spatial transcriptomics for biomedical research," Genome Medicine

---

> > ### Comment · Reviewer_eTFY · 2025-04-01
> >
> > Thank you for submitting rebuttal information, but I think most of my concerns are still not well-resolved, and trajectory infernence is also important for evaluating batch effect correction, as mentioned in scIB. The improvement of this proposed method is also not very interesting, and I think this paper is more suitable for a bioinformatics-specific journals or conferences.

---

> > > ### Author Response · Authors · 2025-04-03
> > >
> > > Thank you for taking the time to review my paper.
> > >
> > > > trajectory infernence is also important for evaluating batch effect correction, as mentioned in scIB
> > >
> > > Please note that as per the reviewer’s request, [in Q5-1 of our first rebuttal](https://openreview.net/forum?id=jeJGH6UDOL&noteId=yvHFdm7jgv), we have indeed conducted trajectory inference after integrating two slices (Refer to Figure 3 and 4 in the [external link](https://anonymous.4open.science/r/Spotscape-31B6/Rebuttal.pdf)). While we are not entirely certain why it may have seemed that trajectory inference was not addressed in our initial rebuttal, your comments suggest that you may have been referring to the "trajectory conservation" metric used in scIB.
> > >
> > > We would like to highlight the difference between our reported correlation and the "trajectory conservation" metric in scIB: we used Pearson correlation, whereas scIB employed Spearman correlation with scaling. In response to this, we have now included the "trajectory conservation" metric in Table 3 of the external link. The updated results are presented below.
> > >
> > > |   | Trajectory conservation  |
> > > |----|-----:|
> > > | Raw   | 0.27 (0.00)   |
> > > | GraphST | 0.16 (0.14)  |
> > > | STAligner | 0.35 (0.26) |
> > > | CAST  | 0.26 (0.45) |
> > > | **Spotscape** | 0.97 (0.02) |
> > >
> > > The updated results demonstrate that our method performs well in terms of conserving the biological signal related to the trajectory.
> > >
> > > > I think this paper is more suitable for a bioinformatics-specific journals or conferences
> > >
> > > We would like to point out that the [ICML 2025 call for papers](https://icml.cc/Conferences/2025/CallForPapers) explicitly highlights "Application-Driven Machine Learning" as one of their topics of interest, with "biosciences" mentioned as an example.
> > > In fact, research on ‘Spatially Resolved Gene Expression’ has also been published in NeurIPS (https://arxiv.org/pdf/2306.01859) and ICLR (https://arxiv.org/pdf/2501.15598), both of which are highly regarded venues with a similar scope and focus to ICML.
> > > Furthermore, "AI for Science" has emerged as a highly prominent and actively researched topic within leading computer science conferences. Therefore, we would like to respectfully emphasize that this topic is not only suitable for bioinformatics journals and conferences, but also highly relevant to broader machine learning venues.
> > >
> > > > The improvement of this proposed method is also not very interesting
> > >
> > > While we are not certain why our work may not have fully captured your interest, we would like to respectfully highlight that, in contrast to more established areas within computer science such as computer vision, the application of AI to spatial transcriptomics remains in its early stages and offers significant potential for impactful contributions. This area requires simple yet practical solutions. Our proposed method is the first to address a variety of downstream tasks with strong performance, even offering fast inference times.
> > >
> > > If the reason for your lack of interest is due to unresolved concerns in other questions, we would like to elaborate further on Q1–Q4. However, due to character restrictions, we will address these briefly:
> > >
> > > **Q1) Presentation of Figure 1** : We believe we have provided a more detailed explanation of this figure. I would like to emphasize once again that the performance of GAE with oracle edges (b-3) is based on a synthetic perfect graph, which fully reflects local information. This setting is not realistic, and it serves to highlight our contribution that global relationships are also important. This is why our method performs slightly lower than this setup in terms of Total CA.
> > >
> > > ---
> > >
> > > **Q2) Cell-Type Specific Signals**: We believe this concern is well addressed through the additional experiments presented in Tables 5 and 6 in the external link.
> > >
> > > | Spatial Domain | ARI | NMI | CA |
> > > |----|:----:|:----:|:----:|
> > > | Spotscape (w/ MLP encoder) | 0.20 (0.01) | 0.30 (0.01) | 0.42 (0.02) |
> > > | **Spotscape** | 0.48 (0.02) | 0.64 (0.01) | 0.61 (0.02) |
> > >
> > > | Cell-type | ARI | NMI | CA |
> > > |----|:----:|:----:|:----:|
> > > | Spotscape (w/ MLP encoder) | 0.58 (0.03) | 0.65 (0.02) | 0.67 (0.03) |
> > > | **Spotscape** | 0.61 (0.07) | 0.68 (0.03) | 0.74 (0.06) |
> > >
> > > ---
> > >
> > > **Q3) Scalability**: This question mainly concerns running time, which we have clearly explained.
> > >
> > > | # of slices (# of spots) | 2 (71,192) | 5 (222,482) | 10 (475,812) | 20 (941,625) | 39 (1,792,797) |
> > > |--|--:|--:|--:|--:|--:|
> > > | STAligner | 2,054 | 6,084 | 21,015 | 273,929  | OOM |
> > > | **Spotscape** | 2,113 | 3,755 | 12,221 | 18,467 | 35,586 |
> > >
> > > ---
> > >
> > > **Q4-1) Hyperparameter Settings**: These have been thoroughly covered in Appendix E.1 in our first submission.
> > >
> > > **Q4-2) Baselines**: The proposed baselines are included in our first rebuttal.
> > >
> > > ---
> > >
> > > We sincerely hope that all of your concerns have been fully addressed and kindly request that you update the score.

---

### Official Review · Reviewer_uFL9 · 2025-03-11

**Overall Recommendation:** 4

**Summary:**

The paper introduces Spotscape, a novel framework for representation learning in Spatially Resolved Transcriptomics (SRT) data. Spotscape incorporates a Similarity Telescope module to capture global similarity relationships and integrates Prototypical Contrastive Learning (PCL) and a similarity scaling strategy to tackle challenges in both single-slice and multi-slice tasks. Extensive experiments demonstrate Spotscape’s superiority in spatial domain identification, trajectory inference, imputation, and multi-slice integration compared to other baselines.

**Claims And Evidence:**

Most of the paper’s claims are supported by the presented experiments and results; however, there are a few issues:
1. In Section 5.3 on Scalability, the authors discuss only the training time. They should also present evidence of performance improvements, as both efficiency and effectiveness are important when assessing scalability.
2. For imputation tasks, the manuscript lacks comparisons with domain-specific imputation methods. Including results from specialized methods would strengthen this claim.

**Essential References Not Discussed:**

For imputation tasks, the authors should compare Spotscape with specialized imputation methods. Including domain-specific methods would enhance the credibility of their claims and provide a more comprehensive evaluation.

**Experimental Designs Or Analyses:**

The experimental framework is generally sound, with extensive benchmarking and detailed evaluations. However, some aspects could be improved:
1. The results are reported as mean ± standard deviation over 10 runs. Incorporating statistical significance tests (e.g., t-tests) would help confirm that the improvements are statistically robust.
2. It appears that Spotscape requires considerable effort for hyper-parameter tuning. The authors should provide guidance or strategies for tuning these hyper-parameters to benefit readers who may want to reproduce or build upon this work.

**Methods And Evaluation Criteria:**

The methods and evaluation criteria are well designed. The paper provides rigorous benchmarking across diverse datasets and tasks, which supports the evaluation of Spotscape’s capabilities.

**Other Comments Or Suggestions:**

I recommend that the authors:
1. Include statistical significance tests (e.g., t-tests) to support the experimental findings.
2. Provide comparisons with domain-specific imputation methods to strengthen the claims related to imputation.
3. Offer additional discussion on the theoretical underpinnings of the loss functions (e.g., in formula (8)) and clarify any potential limitations of their approach.
4. Include guidance on the hyper-parameter tuning process to assist readers in replicating the experiments.

**Other Strengths And Weaknesses:**

Other strengths:
1. The paper is well-organized and easy to follow.
2. Although the methodological contribution is incremental, the practical application in SRT data analysis is compelling.
3. The experiments are well-designed, with rigorous benchmarking across multiple datasets and tasks.

Other weaknesses:
1. The scalability analysis is incomplete, as it focuses solely on training time without addressing performance improvements.
2. The manuscript does not compare Spotscape with specialized imputation methods, which is critical for validating the imputation task claims.
3. The paper could benefit from additional statistical analysis and clearer guidance on hyper-parameter tuning.

**Questions For Authors:**

See above.

**Relation To Broader Scientific Literature:**

The paper’s contributions address current limitations in spatially resolved transcriptomics (SRT) analysis, graph-based representation learning, and prototypical contrastive learning for biological data. Specifically, the work builds on:\
(1) Spatially resolved transcriptomics (baselines): STAGATE[1], SpaGCN[2], SpaceFlow[3], etc.\
(2) Graph-based representation learning (methodology): Graph Convolutional Networks (GCN)[4], etc.\
(3) Prototypical contrastive learning (methodology): scGPCL[5], scPoli[6], etc.

[1] Deciphering spatial domains from spatially resolved transcriptomics with an adaptive graph attention auto-encoder[J]. Nature communications, 2022.\
[2] SpaGCN: Integrating gene expression, spatial location and histology to identify spatial domains and spatially variable genes by graph convolutional network[J]. Nature methods, 2021.\
[3] Identifying multicellular spatiotemporal organization of cells with SpaceFlow[J]. Nature communications, 2022.\
[4] Semi-supervised classification with graph convolutional networks[J]. 2016.\
[5] Deep single-cell RNA-seq data clustering with graph prototypical contrastive learning[J]. Bioinformatics, 2023.\
[6] Population-level integration of single-cell datasets enables multi-scale analysis across samples[J]. Nature methods, 2023.

**Theoretical Claims:**

The submission does not introduce new theoretical claims or detailed proofs. Instead, the authors justify their methodological contributions through empirical experiments and biological reasoning. It would be beneficial for the authors to discuss any theoretical limitations or provide additional analysis—for instance, by citing references that explain the rationale behind the design of the loss functions in formula (8).

---

> ### Author Rebuttal · Authors · 2025-04-01
>
> Thank you for your positive feedback regarding our manuscript. To address your concerns, we have added tables and figures in this [external link](https://anonymous.4open.science/r/Spotscape-31B6/Rebuttal.pdf).
>
> **W1) The scalability analysis is incomplete**
>
> In our _manuscript_, we initially focused on training time in _Figure 10_ because our performance improvements had already been demonstrated through various experiments. The scalability experiment was conducted using synthetic data without meaningful spot information, making it unsuitable for reporting performance metrics.
>
> To provide a more comprehensive evaluation, we generated another synthetic dataset using scCube [1], with mouse embryo data as the reference. In **Figure 6 in the external link**, we demonstrate that Spotscape not only achieves fast inference but also maintains robust performance across varying numbers of spots and slices.
>
> **Q1) t-tests results**
>
> In the **external link**, we have conducted t-tests for all our experiments. We indicate statistical significance with ** for p-value < 0.01 and * for p-value < 0.05.
>
> **Q2) Comparison with domain-specific imputation methods**
>
> Most existing imputation methods in spatial transcriptomics, such as Tangram [2] and stDiff [3], rely on single-cell RNA sequencing data as a reference for improvement. However, this approach is not always applicable, as single-cell reference data is not always available, making these methods unsuitable as direct baselines for our study.
>
> Given this limitation, we considered STAGATE, GraphST, SpaCAE, and SEDR as state-of-the-art baselines, as they denoise input expression data using decoded outputs and have also reported imputation results in their respective papers.
> To further address your concern, we additionally include stMCDI [4], which employs masked conditional diffusion strategies. To our knowledge, this is the most recent baseline that operates under the same setting as ours. Through this experiment in **Figure 7 (external link)**, we demonstrate that Spotscape continues to achieve the best results.
>
> **Q3) Discussion on the theoretical underpinnings of the loss functions**
>
> As you pointed out in your review, our paper primarily focused on practical applications rather than theoretical analysis. However, to provide more clarity on the theoretical underpinnings of our approach, we would like to elaborate on the similarity consistency loss presented in _Equation (1) of our manuscript_. This loss function is a key contribution and one of the main components in _Equation (8) of our manuscript_. We discussed this loss in our rebuttal regarding **Q1** for reviewer **aaPZ** due to character limits.
>
> Meanwhile, outliers may potentially impact the embeddings of other nodes, as the loss considers global relationships, which could be a limitation. Despite these challenges, the consistency loss remains an effective tool for improving the model's global contextual understanding.
>
> **Q4) Guidance on the hyper-parameter tuning process**
>
> _In lines 245-253 of our manuscript_, we clarify that
> > To ensure fairness, we conduct a hyperparameter search for all baseline methods instead of using their default settings, as optimal hyperparameters may vary across datasets. The best-performing hyperparameters are determined based on NMI using the first seed. For Spotscape, only the learning rate is searched using the same criterion. Details of the selected hyperparameters and search spaces are provided in _Appendix E.1 of our manuscript_.
>
> We ensure fairness in the comparison by conducting a broader hyperparameter search for the baseline methods compared to our proposed method, making the comparison as fair and consistent as possible. Specifically, _in Appendix E.1 of our manuscript_, we state that for Spotscape, we utilized default parameters across all datasets except for the learning rate. However, for the baseline methods, additional parameters, such as loss balancing, are also tuned along with the learning rate because some baselines are highly sensitive to them. In this section, we also provide the full search spaces for each baseline.
> To further address your concern, we have now added the final selected hyperparameters for all baseline methods across all datasets to our [Anonymous Github](https://anonymous.4open.science/r/Spotscape-31B6/config/README.md), ensuring full transparency and reproducibility.
>
> ---
> [1] "Simulating multiple variability in spatially resolved transcriptomics with scCube," Nature Communnications
> [2] "Deep learning and alignment of spatially resolved single-cell transcriptomes with Tangram," Nature Methods
> [3] "stDiff: A diffusion model for imputing spatial transcriptomics through single-cell transcriptomic," Briefings in Bioinformatics
> [4] "stMCDI: Masked Conditional Diffusion Model with Graph Neural Network for Spatial Transcriptomics Data Imputation," BIBM 2024

---

### Decision · Program_Chairs · 2025-05-01

**Decision:**

Accept (poster)

**Comment:**

This paper received mixed recommendations: one Weak Reject (eTFY), one Weak Accept (aapZ), and one Accept (uFL9). The two reviewers with positive ratings explicitly acknowledged the novelty of the proposed framework, the thorough experimental evaluation, and the quality of the presentation. Reviewer eTFY, who rated the paper as Weak Reject, raised concerns about the clarity of Figure 1, the scalability of the approach to large-scale datasets, and the experimental details required for fair comparison. After examining the detailed reviewer feedback and the anonymous external material provided, AC believes that most of these concerns can be adequately addressed.

Reviewer eTFY also commented that the improvements presented in the paper may not be of strong interest to the broader machine learning community. However, given the growing importance of SRT, AC believes that this work could be very valuable to application-oriented researchers.

In summary, based on the assessments of all three reviewers, AC finds that the strengths and contributions of the paper outweigh the concerns, and recommends that the paper be accepted for presentation at ICML 2025. AC also encourages the authors to appropriately incorporate the supporting materials into the final version of the paper.